

**The effects of carbon turnover time on terrestrial ecosystem carbon storage**
Yaner Yan[1], Xuhui Zhou[2*], Lifen Jiang[3], Yiqi Luo[3, 4]
[1] Key Laboratory for Eco-Agricultural Biotechnology around Hongze Lake/Collaborative Innovation
Center of Regional Modern Agriculture & Environmental Protection, Huaiyin Normal University,
Huai'an 223300, China
[2]State Key Laboratory of Estuarine and Coastal Research, Center for Global Change and Ecological
forecasting, Tiantong National Field Observation Station for Forest Ecosystem, School of Ecological
and Environmental Sciences, East China Normal University, Shanghai 200062, China
[3]Department of Microbiology and Plant Biology, University of Oklahoma, OK, USA
[4]Center for Earth System Science, Tsinghua University, Beijing, China
Correspondence to: Xuhui Zhou (xhzhou@des.ecnu.edu.cn)

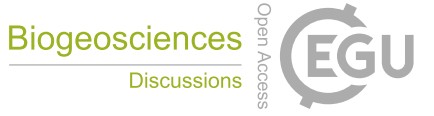

**Abstract.** Carbon (C) turnover time is a key factor in determining C storage capacity in various plant

and soil pools and the magnitude of terrestrial C sink in a changing climate. However, the effects of C

turnover time on C storage have not been well quantified for previous researches. Here, we first

analyzed the difference among different definition of mean turnover time (MTT) including ecosystem

MTT(MTT$_{EC}$) and soil MTT (MTT$_{soil}$) and its variability in MTT to climate changes, and then evaluated

the changes of ecosystem C storage driven by MTT changes. Our results showed that total GPP-based

ecosystem MTT (MTT$_{EC\_GPP}$:25.0±2.7 years) was shorter than soil MTT (35.5 ±1.2years) and NPP-

based ecosystem MTT (MTT$_{EC\_NPP}$:50.8±3 years) ($MTT_{EC\_GPP}= Cpool/GPP$ &

$MTT_{soil}= Csoil/NPP$ & $MTT_{EC\_NPP}= Cpool/NPP$ , Cpool and Csoil referring as the ecosystem or soil

carbon storage, respectively). At the biome scale, temperature is still the predictor for MTT$_{EC}$ ($R^2$ =

0.77, p<0.001) and MTT$_{soil}$ ($R^2$ = 0.68, p<0.001). There is no clear improvement in the performance of

MTT$_{EC}$ predication when incorporating precipitation into the model ($R^2$ = 0.76, p<0.001). Thus, MTT

decreased by approximately 4 years from 1901 to 2011 when temperature just was considered, resulting

in a large C release from terrestrial ecosystems. The resultant terrestrial C release driven by MTT

decrease only accounted for about 13.5% of than driven by NPP increase (159.3 ± 1.45 vs 1215.4 ±

11.0Pg C) due to the diffidence between both of the product factor (NPP ∗ ΔMTT vs MTT ∗ ΔNPP).

Therefore, the larger uncertainties in the spatial variation of MTT than temporal changes would lead in

a greater impact on ecosystem C storage from spatial pattern of MTT, which may need to be focused on

in the future research.



**Key words:** ecosystem, mean turnover time, MAT, MAP, biome scale



## 1 Introduction

Rising atmospheric $CO_2$ concentrations and the resultant climatic warming can substantially impact the
global carbon (C) budget (IPCC, 2007), leading to a positive or negative feedback to global climate
change (e.g.,Friedlingstein *et al*., 2006; Heimann and Reichstein, 2008). Projections of earth system
models (ESMs) show a substantial decrease in terrestrial C storage as the world warms (Friedlingstein
*et al*., 2006), but the decreased magnitude is difficult to quantify due to the complexity of terrestrial
ecosystems in response to global change, such as forest dieback (Cox *et al*., 2004), storms (Chambers
and Li, 2007), and land use change (Strassmann *et al*., 2008). For example, experimental and modeling
studies generally showed that elevated $CO_2$ would enhance NPP and terrestrial C storage (Nemani *et al*.,
2003; Norby *et al*., 2005), but warming may increase soil respiration rates, contributing to reduced C
storage, especially in the colder regions (Atkin and Tjoelker, 2003; Karhu *et al*., 2014). Therefore, the
response of terrestrial C storage to climate depends on the response of C influx and how C residence
time change in various C pools (i.e., plant, litter and soil pools) (Luo *et al*., 2003; Xia *et al*., 2013) as
reflected in most of the biogeochemical models (Parton *et al*., 1987; Potter *et al*., 1993). Todd-Brown *et*
*al*. (2013) validated soil C simulations from CMIP5 earth system models and found that global soil
carbon varied 5.9 fold across models in response to a 2.6-fold variation in NPP and a 3.6-fold variation
in global soil carbon turnover times. Thus it is key to quantify the time that carbon resides in terrestrial
ecosystems and its relationships with climate, and then the resultant change of terrestrial ecosystem C
storage driven by turnover time changes.



In a given environmental condition, the ecosystem C storage capacity refers to the amount of amount of
C that a terrestrial ecosystem can store at the steady state,    determined by C influx and turnover time
(Xia *et al*., 2013). External environmental forces, such as climate change and land use change, would
dynamically influence both ecosystem C influx and turnover time, and then change terrestrial C storage
capacity. Thus, the changed magnitude of ecosystem C storage sink can be expressed by changes in both
NPP and mean C turnover time. The spatial variation of NPP changes and the effects of climate change
have been relatively well quantified by manipulative experiments (Rustad *et al*., 2001; Luo *et al*., 2006),
satellite data (Zhao and Running, 2010), and data assimilation (Luo *et al*., 2003; Zhou and Luo, 2008;
Zhou *et al*., 2012). Todd-Brown *et al*. (2013) also found that differences in NPP contributed
significantly to differences in soil carbon across models using a reduced complexity model dependent
on NPP and temperature. In contrast, the spatial variation of C turnover time have not well been
quantified due to limited data, especially at regional or global scales.
Ecosystem C turnover time is the average time that a C atom stays in an ecosystem from entrance to the
exit (Barrett, 2002). Several methods have been used to estimate the C turnover time: C balance method
by estimating ratios of C pools and fluxes (Vogt *et al*., 1995), C isotope tracing (Ciais *et al*., 1999;
Randerson *et al*., 1999), and measurements of radiocarbon accumulation in the undisturbed soils
(Trumbore *et al*., 1996). However, most methods mainly focused on various pools (i.e., leaf, root, soil)
and small scale (i.e. C isotope tracing, radiocarbon). The turnover time at region or global scale are



often calculated with the ratio of ratios of C storage to flux, such as soil C turnover time (Gill and
Jackson, 2000; Chen *et al*., 2013). Although there are many estimates of global C turnover time, those
global C turnover time focused on soil C. Spatial distribute of ecosystem C turnover time is relatively
difficult to be estimated (Zhou and Luo, 2008), which needs to incorporate individual plant and soil
pools and their C turnover time into ecosystem models. The inverse modeling has been used to estimate
ecosystem mean C turnover time in USA and Australia (Barrett, 2002; Zhou and Luo, 2008; Zhou *et al*.,
2012). Carvalhais et al. (2014) have estimated ecosystem turnover time as the ratio of carbon storage
(soil and vegetation C) and influxes and the correlation to climate, which focused on the validation of
model-based turnover time and the qualitative relationship with climate. Thompson and Randerson *et al*
(1999) has indicated that there were two types of mean C turnover times for terrestrial ecosystems: the
GPP-based or the NPP-based mean turnover time according to the terrestrial C models for some models
use NPP as their C input and others use just GPP from atmosphere (i.e., NPP is GPP minus autotrophic
respiration). However, there was no clear distinction in most pervious researches. For example, Zhou
and Luo (2008) and Zhou *et al*. (2012) estimated mean turnover time as the NPP-based one. In most of
previous researches, soil turnover time are usually estimated using field sampling as the global turnover
time for model validation. However, the difference between different turnover time definitions was still
not quantified. Therefore, we considered vegetation and litter C data into soil C to extend the global
turnover time and then examined the difference between both. Finally, we focused on the effects of
turnover time on ecosystem C storage with the climate changes.



Thus, this study was designed to quantify the global pattern of ecosystem mean turnover time and its
effects on ecosystem C storage driven by turnover time changes. Meanwhile, we also quantified the
difference between different definitions of turnover time. Ecosystem mean turnover time was estimated
using the C balance method, which are ratios of C pools and fluxes. Ecosystem C pools include plant,
litter and soil, and C fluxes refer to ecosystem respiration or C influx (GPP/NPP). The current datasets
from published or unpublished papers have covered all C pools and fluxes, but they were at different
spatial scales, so we estimated ecosystem mean turnover time at the grid (1°×1°) and biome scale for
accuracy and data match. Our objectives are: 1) to estimate the different between ecosystem and soil
mean turnover time, 2) to explore their relationships with climate, and 2) to quantify the ecosystem C
storage changes driven by ecosystem turnover time from 1901 to 2011.
**2   Materials and methods**
2.1 Data collections
Three datasets were used to calculate ecosystem mean turnover time and its climate effects on C
sequestration, including carbon (C) influx (GPP and NPP), C storage in C pools (soil, plant and litter),
and climate factors (temperature, precipitation and potential evapotranspiration). GPP and NPP were
extracted from MODIS products (MOD17) on an 8-day interval with a nominal 1-km resolution since
Feb. 24, 2000. The multi-annual average GPP/NPP from 2000-2009 with the spatial resolution of 0.083°
×0.083° were used in this study (Zhao and Running, 2010).



The harmonized World Soil Database (HWSD) (Hiederer and Köchy, 2012) provided empirical
estimates of global soil C storage, a product of the Food and Agriculture Organization of the United
Nations and the Land Use Change and Agriculture Program of the International Institute for Applied
System Analysis (FAO/IIASA/ISRIC/ISSCAS/JRC, 2012). Hiederer and Köchy (2012) estimated global
soil organic carbon (SOC) at the topsoil (0-30cm) and the subsoil layer (30-100cm) from the amended
HWSD with estimates derived from other global datasets for these layers. We used the amended HWSD
SOC to calculate C turnover time (http://eusoils.jrc.ec.europa.eu). However, HWSD just only provided
an estimate of soil carbon C storage at the top 1 m of soil and may have largely underestimated total soil
carbon. Jobbagy and Jackson (2000) indicated that global SOC storage in the top 3m of soil was 56%
more than that for the first meter, which could change estimates of the turnover time estimates
dramatically. We will discuss this issue in the discussion section. It is well known that HWSD
underestimated soil C in high latitude, so we also estimated turnover time in high latitudes with the
Northern Circumpolar Soil Carbon Database (NCSCD), which is an independent survey of soil carbon
in this region (Tarnocai *et al*., 2009). For biomass, Gibbs (2006) estimated the spatial distribution of the
above- and below-ground C stored in living plant material by updating the classic study (Olson *et al*.,
1983; Olson *et al*., 1985) with a contemporary map of global vegetation distribution (Global Land
Cover database)(Bartholomé and Belward, 2005). Each cell in the gridded data set was coded with an
estimate of mean and maximum carbon density values based upon its land cover class, so this dataset
mainly represents plant biomass C at a biome level.



The litter dataset was extracted from 650 published and unpublished documents (Holland *et al*., 2005).
Each record represents a site, including site description, method, litterfall, litter mass and nutrients. We
calculated the mean and median of litter mass for each biome, and then assigned the value for each grid
according as the biome types, forming the global pattern of litter C storage using the method of
Matthews (1997) in ARCGIS software.
Global climate databases produced by the Climate Research Unit (CRU) at the University of East
Anglia were used to analyze the climatic effect on ecosystem mean turnover time. We used mean 0.5
$^{o} \times 0.5^{o}$ gridded air temperature, precipitation and potential evapotranspiration, specifically their means
from 2000-2009 in CRU_TS 3.20 (Harris *et al*., 2013).
We aggregated all datasets into a biome level for accuracy and data match, so the biome map was
extracted from the GLC 2000 (Bartholomé and Belward, 2005) and regulated by MODIS. We assigned
22 land cover class among three temperature zones (i.e., tropical, temperate and boreal) by taking the
most common land cover from the original underlying $0.083^{o} \times 0.083^{o}$ data. Eight typical biomes were
zoned with ARCGIS 10 in corresponding to plant function types (PFTs) in CABLE model that Xia *et al*
(2013): evergreen needleleaf forest (ENF), evergreen broadleaf forest (EBF), deciduous needleleaf
forest (DNF), deciduous broadleaf forest (DBF), tundra, shrubland, grassland and cropland. All of the
data were regridded using R software to a common projection (WGS 84) and $1^{0} \times 1^{0}$ spatial resolution.
The regridding approach for C fluxes and pools (i.e., GPP, NPP, soil C and litter C) assumed



conservation of mass that a latitudinal degree was proportional to distance for the close grid cells (Todd-
Brown *et al*., 2013). A nearest neighbor approach were used for land cover classes and a bi-linear
interpolation were used for climate variables (i.e, temperature, precipitation).
2.2 Estimation of ecosystem mean C turnover time
C turnover time is commonly estimated with the C balance method by calculating the ratio of C total in
a C pool and its outflux. Terrestrial ecosystem includes many C pools with largely varying residence
times from days to millennia, but it is difficult to collect the observation-based datasets of C pools and
flux for each component (e.g. leaf, wood and different soil C fractions) at the global scale. It thus is
impossible to estimate individual pools' turnover time. In this study, we estimated the whole-ecosystem
C turnover time as the ratio of C pools to flux based on the observed datasets. Certainly, there are some
limitations that the ecosystem is taken as a single pool, which will be discussed in the discussion. For
terrestrial ecosystems, the C pools ($C_{pool}$) is composed of three parts: plant, litter and soil, and C
outfluxes include all C losses include autotrophic and heterotrophic respiration ($R_a$, $R_h$) and losses by
fires and harvest. In the steady state, C outfluxes equal to C influx, which is the carbon uptake through
gross primary production (GPP), so ecosystem mean turnover time ($MTT_{EC}$) can be equivalently
calculated as the ratio between C storage in vegetation, soils and litters, and the influx into the pools,
GPP:



$$MTT_{EC} = \frac{Cpool}{GPP} \qquad (1)$$

The similar method was used to calculate soil MTT ($MTT_{soil}$):

$$MTT_{soil} = \frac{Csoil}{NPP} \qquad (2)$$

However, the steady-state in nature is rare, so we relax the strict steady-state assumption and computed
the ratio of $C_{pool}$ to GPP as apparent whole-ecosystem turnover time and interpret the quantity as an
emergent diagnostic at ecosystem level (Carvalhais *et al*., 2014). In addition, it is difficult to accurately
get the observed respiration ($R_a$ and $R_h$) in terrestrial ecosystem at the global scale. Therefore, we used
multi-year GPP or NPP to calculate MTT in order to reduce the effect of the non-steady state, since it is
difficult to evaluate how this assumption affects model results.

2.3 The climate effects on ecosystem mean C turnover time
In order to explore the combing effect of precipitation and temperature on ecosystem mean C turnover
time, aridity index (AI) was calculated as follows:

$$AI = \frac{MAP}{PET} \qquad (3)$$

where PET is the potential evapotranspiration and MAP is mean annual precipitation (Middleton and





Thomas, 1997). AI is a bioclimatic index including both physical phenomena (precipitation and
potential evapotranspiration) and biological processes (plant transpiration) related with edaphic factors.
The relationships were examined between ecosystem mean C turnover time and mean annual
temperature (MAT, $^{o}$C), mean annual precipitation (MAP, mm) and aridity index (AI) at the biome level.
The regression analyses  $(MTT = ae^{-bMAT/MAP})$were performed in STATISTICA 10, where a and b are
the coefficients. The coefficient of determination ($R^2$) was used to measure the phase correlation
between ecosystem mean C turnover time and climate factors. Here, we calculated a $Q_{10}$ value (i.e., $Q_{10}$,
a relative increase in mean turnover time for a 10$^{o}$C increase in temperature, $Q_{10} = e^{10b}$, b, the
coefficients of $MTT = ae^{-bMAT/MAP}$) that is used in most models to simulate C decomposition. The
relationship between ecosystem mean turnover time and temperature was used to estimate mean C
turnover time in 1901 and 2011.
2.4 The effects of turnover time on ecosystem C storage
Ecosystem C storage capacity at steady state is represented by$NPP \times MTT$  (Lou *et al*., 2003), so the
difference of ecosystem C storage from 1901 to 2011 can be calculated as follows:

$$\Delta Cpool = NPP_{2011} \times MTT_{2011} - NPP_{1901} \times MTT_{1901}$$

$$\Rightarrow \Delta Cpool = NPP_{2011} \times MTT_{2011} - (NPP_{2011} - \Delta NPP) \times (MTT_{2011} - \Delta MRT) \qquad (4)$$

$$\Rightarrow \Delta Cpool = NPP_{2011} \times \Delta MTT + MTT_{2011} \times \Delta NPP - \Delta NPP \times \Delta MTT$$



where $NPP_{1901(2011)}$ and $MTT_{1901(2011)}$ refer to NPP and MTT at time 1901 or 2011. $\Delta Cpool$ ($\Delta NPP$ or
$\Delta MTT$) is the difference between ecosystem C storage (NPP or MTT) at time 2011 and that at time
1901. The first component ($NPP_{2011} \times \Delta MTT$) represents the effects of MTT changes on ecosystem C
storage. The second component ($\Delta NPP \times MTT_{2011}$) is the effects of NPP change on ecosystem C storage,
and $\Delta NPP \times \Delta MTT$ is the cross-coupling effects.
To assess the effects of changes in MTT or NPP on ecosystem C storage, ecosystem MTT in 1901 and
2011 was calculated using an exponential equation between mean turnover time and temperature at a
biome level. NPP in 2011 was derived from products (MOD17) and NPP in 1901 was averaged from the
eight models' simulated results (CanESM2, CCSM4, IPSL-CM5A-LR, IPSL-CM5B-LR, MIROC-
ESM, MIROC-ESM-CHEM, NorESM1-M and NorESM1-ME) for modeled NPP is near to MODIS
estimated NPP (Yan *et al*., 2014).
2.5 Uncertainty analysis and sensitivity Analysis
Limitation of the above datasets is that the uncertainties are poorly quantified. The global mean of C
fluxes (GPP and NPP) and pools (soil, litter, and plant) were calculated by 1000 simulations,
respectively, through Markov chain Monte Carlo (MCMC) sampling from a gamma distribution in R
software. For each variable, the confidence interval (CI) was estimated as the 2.5 and 97.5 percentile of
mean values of the 1000 simulations. It was also applied to estimate the confidence interval of



ecosystem C storage and ecosystem mean C turnover time.

## 3    Results

3.1 Ecosystem C storage
On average, terrestrial C storage (plant biomass + soil + litter) was 22.0 kg C m$^{-2}$ (with a 95% CI of
21.85- 22.50 kg C m$^{-2}$) at the global scale, which largely varied with vegetation and soil types (Fig.1d).
Among the forest biomes, ecosystem C storage was highest in boreal evergreen needleaf forest (ENF)
with high soil C content and lowest in deciduous broadleaf forest (DBF) with the lowest soil C. Soil C
was the largest C pool in terrestrial ecosystems, accounting for more than 60% of ecosystem C storage,
while C storages in litter and biomass only represented less than 10% and 30%, respectively (Fig. 1b).
Among eight typical biomes associated with plant functional types (PFTs) (Table 1), the order of
ecosystem C storage followed as: ENF (34.84±0.02 kg C m$^{-2}$) > deciduous needleleaf forest (DNF,
25.30±0.03 kg C m$^{-2}$)> evergreen broadleaf forest (EBF, 22.70±0.01 kg C m$^{-2}$)> shrubland (18.29±0.02
kg C m$^{-2}$) > DBF (16.51±0.02 kg C m$^{-2}$) > tundra (14.16 ±0.02 kg C m$^{-2}$)/cropland (14.58 ±0.01kg C m$^{-}$
$^{2}$)> grassland(10.80±0.01 kg C m$^{-2}$).
3.2 Mean C turnover time
On average, ecosystem mean C turnover time (MTT) was 25.0 years (with a 95% CI of 23.3-27.7 years)
based on GPP data and 50.8 years (with a 95% CI of 47.8-53.8 years) on NPP data (Table 1), while soil



MTT is smaller than NPP-based MTT with the value of 35.5 years (with a 95% CI of 34.9-36.7 years).
MTT varies among biomes due to the different climate forcing (Table 1 and Fig 2). The long MTT
occurred in high latitude while the short ones are in tropical zone. Among forest biomes, DNF had the
highest MTT with the lowest mean temperature (-7.9 $^{o}$C), while the lowest MTT was in EBF due to
highest temperature (24.5 $^{o}$C) and precipitation (2143 mm). Although ecosystem C storage was low in
tundra (14.16 kg C m$^{-2}$), it has the longest MTT. Therefore, the order of ecosystem MTT among biomes
was different from that of ecosystem C storage, with tundra (99.704±6.14 years) > DNF (45.27±
2.43years) or ENF (42.23±2.01 years) > shrubland (27.77±2.25 years) > grassland (26.00±1.41 years) >
cropland (14.91±0.40years) or DBF (13.29± 0.68years) > EBF (9.67±0.21 years). Soil MTT had the
similar order with ecosystem MTT with the different values (Table 1). In the high latitude, ecosystem
MTT could increase up to 145 years if soil C storage was calculated from NCSCD dataset (Fig. 3) due
to higher soil C storage (500 Pg C vs 290 Pg C), compared with the global soil C storage HWSD, while
the global average of soil MTT increased to 40.8 years when NCSCD dataset was considered.
3.3 Climate effects on ecosystem mean turnover time
Ecosystem mean C turnover time significantly decreased with mean annual temperature (MAT) and
mean annual precipitation (MAP) as described by an exponential equation:$MTT = 57.06e^{-0.07MAT}$
(R$^2$=0.77, P<0.001) and $MTT = 103.07e^{-0.001MAP}$ (R$^2$=0.34, P<0.001, Fig 4), but there was no
correlation between ecosystem mean turnover time and aridity index (AI, Fig. 4c). The similar





relationships occurred between soil MTT and MAT and MAP ($MTT_{soil} = 58.40e^{-0.08MAT}$, R$^2$=0.68,
P<0.001) and  $MTT_{soil} = 109.98e^{-0.002MAP}$ , R$^2$=0.48, P<0.001, Fig. 5). There was the different
temperature sensitivity of mean turnover time ($Q_{10}$) for ecosystem MTT ($Q_{10}$=1.95) and soil MTT
($Q_{10}$=2.23) at ecosystem scale, which was estimated as $Q_{10} = e^{10b}$ based on temperature regression
function. When MAP was incorporated into a multivariate regression function of ecosystem mean
turnover time with MAT, the relationships could not be significantly improved (Fig. 6a). While MAP
improved the explanation of variance of soil MTT (R$^2$ from 0.68 to 0.76, Fig. 6b), although there were
the relationships due to the significant covariance of MAP and MAT (R$^2$=0.60). However, the
relationship between MTT and AI is not clear due to the scale limit (biome level). When we separated
ecosystem MTT into two categories according to aridity index (i.e., AI >1 and AI< 1), the relationships
between ecosystem MTT and MAT did not significantly change (Figs. 4e, h) compared to that with all
data together (Fig. 4b), while the relationship of ecosystem MTT with MAP significantly increased
when AI > 1, but decreased when AI <1. However, the same regression function of soil MTT with MAT
largely improved the explanation of the variance when AI>1 (Fig. 5e, $MTT = 58.67e^{-0.08MAT}$,
R$^2$=0.76, P<0.001 ). The relationships between soil MTT and MAP were both improved when AI>1 and
AI<1 (Fig. 5e, h).
3.4 Temporal variations of ecosystem mean turnover time and C storage
The average increase in global air temperature is around 1°C from 1901 to 2011 based on the Climate





Research Unit (CRU) datasets, ranging from -2.5 to 5.9 °C. When the function between ecosystem MTT
and temperature was used to estimate the change in ecosystem mean turnover time (Fig. 4), the average
mean turnover time decreased by approximately 4 years (Fig. 7a). The largest change in ecosystem
MTT occurred in the cold zones. In tundra, mean C turnover time decreased by more than 10 years due
to the larger increase in temperature (~2°C) than other regions. However, the average NPP increased by
approximately $0.3\pm0.003$ Kg C m$^{-2}$ yr$^{-1}$ over 110 years with most range of 0~0.6 Kg C m$^{-2}$ yr$^{-1}$ (Fig. 7b).
The changes in ecosystem MTT and NPP across 110 years would cause decrease or increase in
terrestrial C storage. Driven by MTT changes, ecosystem C storage decreased by $159.3 \pm 1.45$ Pg C yr$^{-1}$
from 1901 to 2011 ($\Delta$MTT $\times$ NPP), with the largest decrease in tundra and boreal forest (more than 12 g
C m$^{-2}$ yr$^{-1}$) but little decrease in tropical zones (Fig. 8a). However, the increase in NPP directly raised
ecosystem C storage up to $1215.4 \pm 11.0$ Pg C yr$^{-1}$ from 1901 to 2011 with a range of 30-150 g C m$^{-2}$ yr$^{-}$
$^1$ in most areas (MTT $\times$ $\Delta$NPP, Fig. 8b). The MTT-induced changes in ecosystem C storage only
accounted for about 13.5% of that driven by NPP due to the difference between both of the product
factor, so the spatial pattern of the NPP-driven changes mostly represented the spatial pattern of the
changes in ecosystem C storage (Fig. 6d).
**4   Discussion**
4.1 Global pattern of mean turnover time
In this study, we estimated spatial patterns of mean turnover time (MTT) with ecosystem C influxes





(GPP and NPP) and C pools in plants, litter and soil using the C balance method. Here, we assumed that
the nature was the steady state and took the whole ecosystem as a single pool similar in Sanderman *et al*
(2003), which have some caveats in the estimation of mean turnover time. Terrestrial ecosystems
comprise of compartments varying greatly in their individual turnover times (for example leaves, wood,
different soil organic carbon fractions), but we cannot estimate turnover time for each pools using
observation datasets. In addition, it is difficult to accurately get the observed respiration ($R_a$ and $R_h$) in
terrestrial ecosystem at the global scale, or carbon allocation between outflux and influx. It is thus
difficult to evaluate how this assumption affects model results. Maybe, inverse models would be a valid
method to estimate turnover time for the both (e.g., Zhou *et al*., 2012).
The global average of ecosystem MTT was 25.0 years for GPP-based estimation and 50.8 years for
NPP-based one and soil MTT was 35.5 years, which was within the global mean turnover times (26-60
years) estimated by various experimental and modeling approaches with NPP-based estimation
(Randerson *et al*., 1999; Thompson and Randerson, 1999) mostly focused on soils, but not ecosystem
MTT. However, our results indicated that ecosystem MTT (GPP-based estimation) was shorter than soil
MTT ($MTT_{EC} = Cpool/GPP$ & $MTT_{soil} = Csoil/NPP$ ). According to the equations, the difference
between ecosystem and soil MTT depends on the component carbon pools and the ratio of GPP to NPP.
Thus, there was subtle difference in patterns of MTT between both. For example, ecosystem MTT in
Evergreen Needleleaf forest (ENF) was larger than soil MTT where the decomposition rate in soil C
was very slow.





The GPP-based MTT were also larger than the result of Carvalhais *et al* (2014) (23 years), probably
due to litter C storage included in this study. The ratio of GPP-based and NPP-based MTT (0.49) was
smaller than that estimated by Thompson and Randerson (1999) (0.58, 15 year vs. 26 year,
respectively). Our NPP-based MTTs for the conterminous USA (37.2 years) and Australia (33.4 years)
were shorter than the estimates by the inverse models (46 to 78 years) (Barrett, 2002; Zhou and Luo,
2008; Zhou *et al*., 2012). The NPP-based MTT was lower than the estimated results from Xia *et al*.
(2013) using the CABLE model, though the order of MTT across forest biomes is similar. In addition,
we only used soil C in the top 1 m to estimate ecosystem MTT, which would be largely underestimated
for the important amounts of C stored between 1 and 3m depth (Jobbagy and Jackson, 2000). According
to the SOC estimation of Jobbagy and Jackson (2000), the MTT in the top 3 m could increase to 34.63
years for GPP-based, 70.68 years for NPP-based and 55.38 years for soil. Therefore, the accurate
estimates of total soil C are important to estimate ecosystem MTT.
4.2 The sensitivity of turnover time to climate
The estimated mean turnover time (MTT) was shortest in tropical zones and increased toward high-
latitude zones (Fig. 2), which were often affected by the spatial patterns of temperature and moisture.
The results was similar to those the previous studies based on SOC data set (Schimel *et al*., 1994;
Sanderman *et al*., 2003; Frank *et al*., 2012; Chen *et al*., 2013) and root pools (Gill and Jackson, 2000).
Ecosystem MTT had negative exponential relationship with MAT (Fig 4), similar to those with soil



MTT, probably due to the temperature dependence of respiration (Lloyd and Taylor, 1994; Wen *et al.*,
2006). Our results showed that the temperature sensitivity of ecosystem MTT was lower than that of
soil C pool ($Q_{10}$: 1.95 vs. 2.23, Figs. 4 &5), which was similar to the previous research (Sanderman *et*
*al.*, 2003), because wood may decompose at much lower rates than SOM due to the longer MTT of
wood (Zhou *et al.*, 2012). Ecosystem MTT was no significant differences between very humid zone
(AI>1.0) and other zones (AI<1.0, Fig 4). However, the better relationships between MTT and MAP
occurred in very humid zone (AI>1.0) than other zones, which was similar to soil pool, but soil MTT
have the higher sensitivity to precipitation than ecosystem MTT under AI>1. SOM decomposition often
increased with added moisture in aerobic soils (Trumbore, 1997), because the metabolic loss of various
C pools increased under warmer and wetter climates (Frank *et al.*, 2012), resulting in high sensitivity of
MTT to MAP. Thus, the fitting regression combined MAT and MAP clearly improved soil MTT
($R^2$=0.76, p<0.001, Fig 6b). In arid or semi-humid regions, the increase in C influx with MAP was more
rapid than that in decomposition (Austin and Sala, 2002). In addition, water limitation may suppress the
effective ecosystem-level response of respiration to temperature (Reichstein et al., 2007). At an annual
scale, temperature is still the best predictor of MTT (Chen et al., 2013), which explained up to 77% of
variation of MTT (Fig 4). Other ecosystem properties (e.g. ecosystems types, soil nitrogen) may cause
the rest of the variation in the estimates of MTT.
4.3 Effects of the changes in mean turnover time on ecosystem C storage



Terrestrial ecosystems play an important role in regulating C cycling balance to combat global change.
Current studies suggest that the terrestrial biosphere is currently a net C sink (Lund *et al.*, 2010), but it
is difficult to assess the sustainability of ecosystem C storage due to the complexity of terrestrial
ecosystem in response to global change (Luo, 2007). In this study, we first tried to assess the potential
shifts of ecosystem C storage capacity by changes in both NPP and ecosystem MTT. Our studies
indicated that the decrease in MTT increased ecosystem C loss over time while increased NPP enhance
ecosystem C uptake.
Current datasets have showed an increase in NPP (e.g., Hicke *et al.*, 2002; Potter *et al.*, 2012), leading
to increasing terrestrial C uptake. Driven by NPP changes from 1901 to 2011, our results showed that
global C storage would increase by 11.0 Pg C yr$^{-1}$ and 0.4 Pg C yr$^{-1}$ at the global scale and conterminous
USA, respectively. Our estimated ecosystem C storage in USA was larger than the one from inverse
models (Zhou and Luo, 2008; Zhou *et al.*, 2012) but comparable to C sink from atmospheric inversion
(0.30-0.58 Pg C yr-1) (Pacala *et al.*, 2001). However, the shortened MTT caused C losses from
ecosystems from 1901 to 2011 (about 1.45 Pg C yr$^{-1}$), indicating that the magnitude of ecosystem C
uptake is likely to decrease under warming due to decreased MTT. Ecosystem C losses driven by the
decrease in MTT only accounted for 13.5% of ecosystem C uptake compared to that driven by NPP
increase, still causing a net sink in terrestrial ecosystem. The largest changes in terrestrial C storage
occurred in high latitude, where it is more vulnerable to loss with climate change (Zimov *et al.*, 2006).
However, the direct release of $CO_2$ in high latitude through thawing would be another large source in



the decrease of ecosystem C storage under climate warming (Grosse *et al.*, 2011), which cannot be
assessed by MTT or NPP. Interestingly, our results suggested that the substantial changes in terrestrial C
storage occurred in forest and shrub (50% of total) due to the relatively longer MTT, which caused the
larger terrestrial C uptake driven by NPP increase compared with others. Moreover, the largest absolute
and relative changes of MTT occurred in high latitude regions (Fig. 7a), which would largely decrease
the terrestrial C uptake driven by NPP under global warming. Furthermore, the C uptake in cropland
and grassland has been underestimated probably due to the ignorance of the effects of land
management.
4.4 Limitation in estimating mean turnover time and its effects to climate
Estimated MTT in this study were based on C influxes (GPP or NPP) and C pools in plants, litter and
soil at the grid scale and can be used to quantify global, regional or biome-specific MTT, which was
very important to evaluate terrestrial C storage. However, the balance method and data limitation may
cause biases to some degree in estimated ecosystem MTT in a few sources. First, we assumed that
ecosystem C cycle is at the steady state, when MTT was estimated. It is difficult to define the steady
state, especially soil C dynamics (Luo and Weng, 2011). Actually, steady state is rare in nature and any
ecosystem process could be only close to reach the steady state in the short time. For example,
permafrost will be thawing both gradually and catastrophically (Schuur *et al.*, 2008). The assumption of
the steady state would cause the overestimation or underestimation of ecosystem MTT (Zhou *et al.*,



2010). Second, MTT was estimated on the basis of C pool and flux measurements, whose uncertainties
in the current datasets of C pools and fluxes would limit the estimated MTT. For example, the
amendments of typological data and bulk density had largely improved the estimates of the SOC storage
from HWSD (1417 PgC) (Hiederer and Köchy, 2012). Soil C storage calculated from NCSCD dataset
would improve the ecosystem MTT in high latitudes (Fig. 3), compared with that from HWSD datasets.
However, it is difficult to quantify the uncertainty in MTT cuased by uncertainties of the pool and flux
datasets due to lack of quantitative uncertainty estimates in these datasets. The calculation of MTT by
the ratio of the pool to flux would reduce these uncertainties associated with the pool and flux data sets
in some degree.
Third, the uncertainties in ecosystem MTT would cause the uncertainties in the relationship between
MAT, MAP and ecosystem MTT. To simplify the calculation, we aggregated all datasets into a biome
level, leading in a fixed parameters across biomes. However, the response magnitude in soil respiration
to warming varied over time and across sites (Rustad et al., 2001; Davidson and Janssens, 2006),
resulting in mutliple temperature response function. MTT for 1901and 2011 were estimated using the
exponential function between mean turnover time and temperature, resulting in underestimation or
overestimation of MTT and the resultant changes on ecosystem C storage.
4.5 Implication for land surface models



First, this study demonstrated that spatial variability of ecosystem mean C turnover time had higher
uncertainties compared to temporal variability, which was mainly caused by the estimation of soil C
storage. Further work should focus on the accurate estimation of soil C storage with numerous
observational data in estimating the spatial patterns of mean C turnover time at regional or global scale.
Land surface model should consider spatial variability of ecosystem mean C turnover time, especially at
high latitude.
Second, there were the inconsistent responses of ecosystem C turnover time to climate variables in the
current global vegetation models (Friend *et al*., 2013). Our results showed that temperature was the best
predictor for ecosystem C turnover time ($R^2 = 0.77$, p<0.001) on annual scale, which declined with
rising temperature. Such temperature relationship with mean C turnover time can be incorporated into
land surface models to improve the forecast of terrestrial climate-C cycle feedback. Third, our results
showed that temperature sensitivity of ecosystem MTT was lower than that of soil C pool while
precipitation was less sensitive to ecosystem turnover time than soil C turnover time with different
effects in very humid zone and arid zone. Now all global carbon cycle models have considered moisture
stress on vegetation, but the incorporation of moisture or precipitation stress into soil decomposition
should be strengthened, especially in high-latitude zones with greater warming and increased
precipitation.



Ecosystem C turnover time is crucial in determining terrestrial C storage capacity, so it is necessary to
quantify ecosystems turnover time and its relationships with climate. We developed global maps of
ecosystem C mean turnover time based on the current datasets from published GPP and C pools in
plant, litter and soil. The average ecosystem mean turnover time at the global scale is 25.0 years with a
range from about 8 years for spare grassland to 120 years for tundra, which is shorter than soil C pool
alone. Our results showed that the temperature sensitivity of ecosystem turnover time was lower than
that of soil C pool ($Q_{10}$: 1.95 vs. 2.23), while the relationship between ecosystem C turnover time and
precipitation under low aridity conditions (AI>1) was much stronger than for all or AI<1 conditions at
biome scale. MTT decreased by approximately 4 years from 1901 to 2011 when temperature just was
considered, resulting in a large C release from terrestrial ecosystems. The resultant terrestrial C release
driven by MTT decrease only accounted for about 13.5% of than driven by NPP increase (159.3 vs
1215.4 Pg C) due to the diffidence between both of the product factor (NPP*ΔMTT vs MTT*ΔNPP).
Therefore, understanding the response of C turnover time to global warming would be important to
assess the sustainability of ecosystem C storage.
**Data availability**
All of the original elevation data used in this study is referenced in Fig 1 of the manuscript and full
citations for data sources are provided.
**Acknowledgements**
This research was financially supported by The Program for Professor of Special Appointment (Eastern



Scholar) at Shanghai Institutions of Higher Learning, 2012 Shanghai Pujiang Program (12PJ1401400),
and "Thousand Young Talents" Program in China (31370489).

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



**Table 1**. The density of ecosystem C storage (Kg C m-2), mean turnover time (MTT, years), mean annual temperature (MAT) and precipitation (MAP) for the eight biomes. Ecosystem MTT were calculated based GPP and NPP, respectively.

| Biome | Ecosystem C storage (kg C m$^{-2}$) | Ecosystem MTT (years) | | Soil MTT(years) | MAT ($^o$C) | MAP (mm) |
|---|---|---|---|---|---|---|
| | | MTT$_{GPP}$ | MTT$_{NPP}$ | | | |
| ENF | 34.8±0.02 | 42.23±2.01 | 58.54±2.16 | 39.62±1.22 | 3.5 | 760.5 |
| EBF | 22.7±0.01 | 9.67±0.21 | 18.43±0.43 | 8.96±0.21 | 24.5 | 2143.5 |
| DNF | 25.3±0.03 | 45.27±2.43 | 75.80±2.71 | 53.50±1.71 | -7.9 | 401.4 |
| DBF | 16.5±0.02 | 13.29±0.68 | 22.02±1.00 | 12.08±0.69 | 16.1 | 988.4 |
| tundra | 14.2±0.02 | 99.74±6.14 | 132.86±4.40 | 122.88±5.54 | -11.1 | 291.1 |
| Shrubland | 18.3±0.02 | 27.77±2.25 | 43.41±2.37 | 36.22±2.01 | 9.3 | 643.6 |
| Grassland | 10.8±0.01 | 26.00±1.41 | 39.51±2.11 | 34.37±2.20 | 9.4 | 605.5 |
| Cropland | 14.6±0.01 | 14.91±0.40 | 23.06±0.84 | 17.72±0.58 | 15.4 | 885.7 |

*ENF: Evergreen Needleleaf forest; EBF: Evergreen Broadleaf forest; DNF: Deciduous Needleleaf forest; DBF: Deciduous Broadleaf forest.



**Figure Caption List**

**Figure 1**. Spatial pattern of soil C (a), biome C (b), litter C (c) and ecosystem C storage (d) at grid scale

$(1^o \times 1^o)$. Unite: Kg C m$^{-2}$. Ecosystem C storage was calculated from biomass, soil and litter C pools.

**Figure 2**. Spatial pattern of mean turnover time (MTT, years), calculated based on biome types and

GPP (a) or NPP (b) and soil (c) using the C balance methods.

**Figure 3.** Spatial pattern of mean turnover time (years) in high latitude. (a) Based on soil C storage

from HWSD data, (b) based on soil C storage from NCSCD data.

**Figure 4**. Relationships between ecosystem mean turnover time (MTT) and multi-annual temperature

(MAT, a), precipitation (MAP, b) at different aridity indexes (AI, c). Each data point stands for average

values of each biome. Biomes were assigned into 62 types according to land cover and three

temperature zones.

**Figure 5**. Relationships between soil mean turnover time (MTTsoil) and multi-annual temperature

(MAT, a), precipitation (MAP, b) at different aridity indexes (AI, c). Each data point stands for average

values of each biome. Biomes were assigned into 62 types according to land cover and three

temperature zones.

**Figur**e 6. Surface fitting between mean turnover time and multi-annual temperature (MAT),

precipitation (MAP) for ecosystem (a) and soil (b).

**Figure 7.** Change values of ecosystem mean ecosystem mean turnover time (MTT, unit: year a) driven

by temperature change and NPP (unit: Kg C m$^{-2}$yr$^{-1}$) from 1901 to 2011. MTT for 1901 and 2011 was



calculated by the temperature-dependence function showing in Fig. 4. NPP in 1901 and 2011 was
derived from models' average and MODIS.
**Figure 8**. Change values of ecosystem carbon storage driven by mean turnover time change
(NPP$_{2011}$×ΔMTT, a), by NPP change (MTT$_{2011}$×ΔNPP, b) and by NPP change and MRT change
(ΔMTT×ΔNPP, c) and total ecosystem C storage changes (d). Unit: g C m$^{-2}$ yr$^{-1}$ ($\Delta C_{pool} = NPP_{2011} \times$
$\Delta MTT + MTT_{2011} \times \Delta NPP - \Delta NPP \times \Delta MTT$).





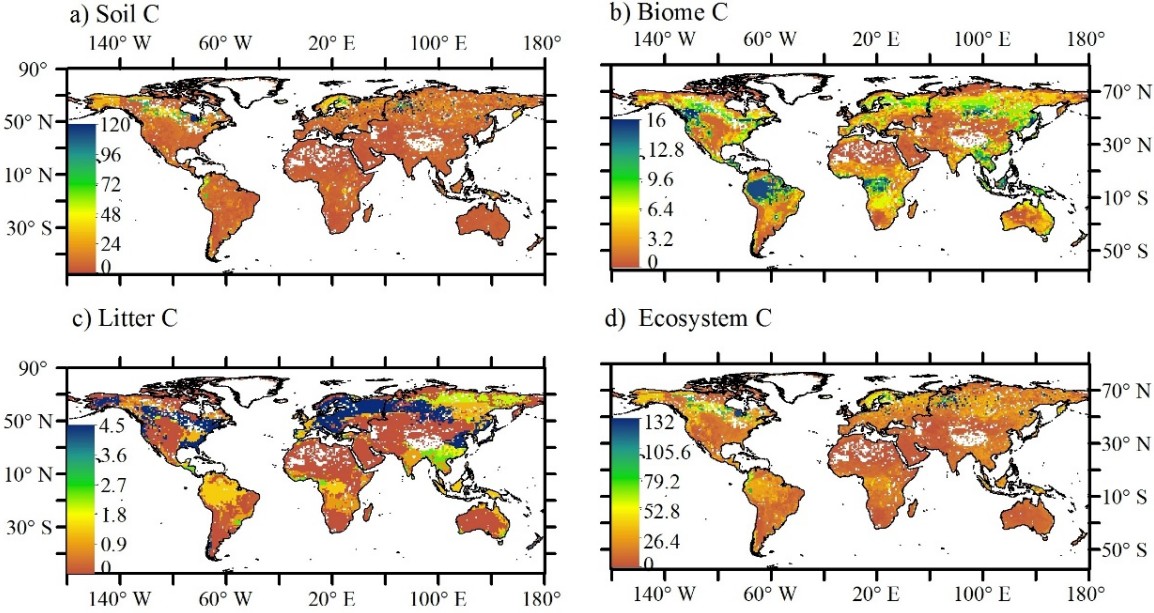

**Figure 1**. Spatial pattern of soil C (a), biome C (b), litter C (c) and ecosystem C storage (d) at grid

scale (1°×1°). Unite: Kg C m$^{-2}$. Ecosystem C storage was calculated from biomass, soil and litter C

pools.




**Figure 2**. Spatial pattern of mean turnover time (MTT, years), calculated based on biome types and

GPP (a) or NPP (b) and soil (c) using the C balance methods.




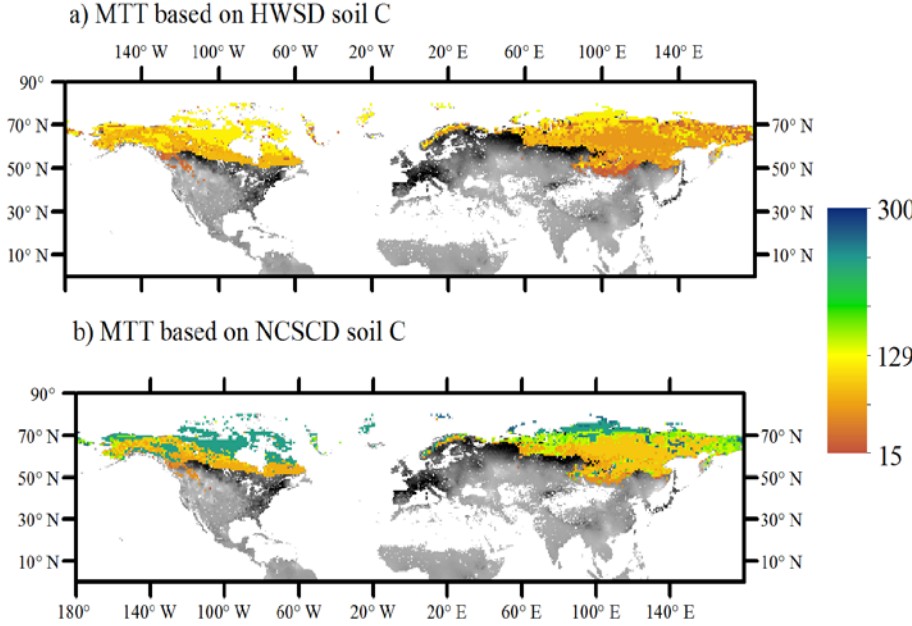

**Figure 3**. Spatial pattern of mean turnover time (years) in high latitude. (a) Based on soil C storage
from HWSD data, (b) based on soil C storage from NCSCD data.







**Figure 4**. Relationships between ecosystem mean turnover time (MTT) and multi-annual

temperature (MAT, a), precipitation (MAP, b) at different aridity indexes (AI, c). Each data point stands

for average values of each biome. Biomes were assigned into 62 types according to land cover and three

temperature zones.

619


620

**Figure 5**. Relationships between soil mean turnover time (MTT$_{soil}$) and multi-annual temperature

(MAT, a), precipitation (MAP, b) at different aridity indexes (AI, c). Each data point stands for average

values of each biome. Biomes were assigned into 62 types according to land cover and three

temperature zones.





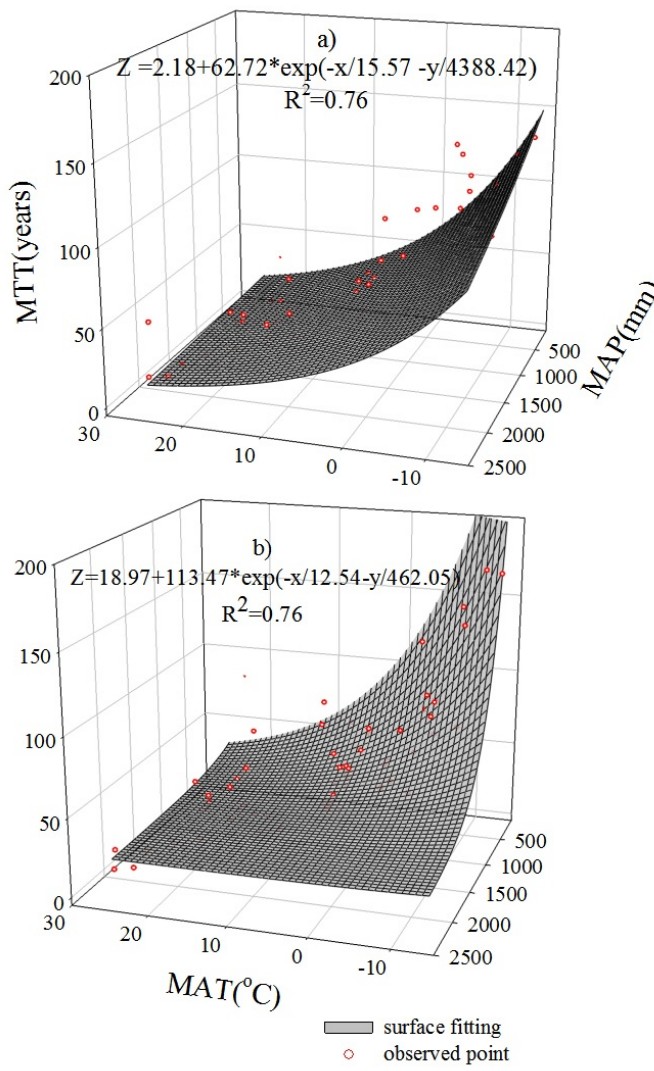

**Figure 6**. Surface fitting between mean turnover time and multi-annual temperature (MAT),

precipitation (MAP) for ecosystem (a) and soil (b).





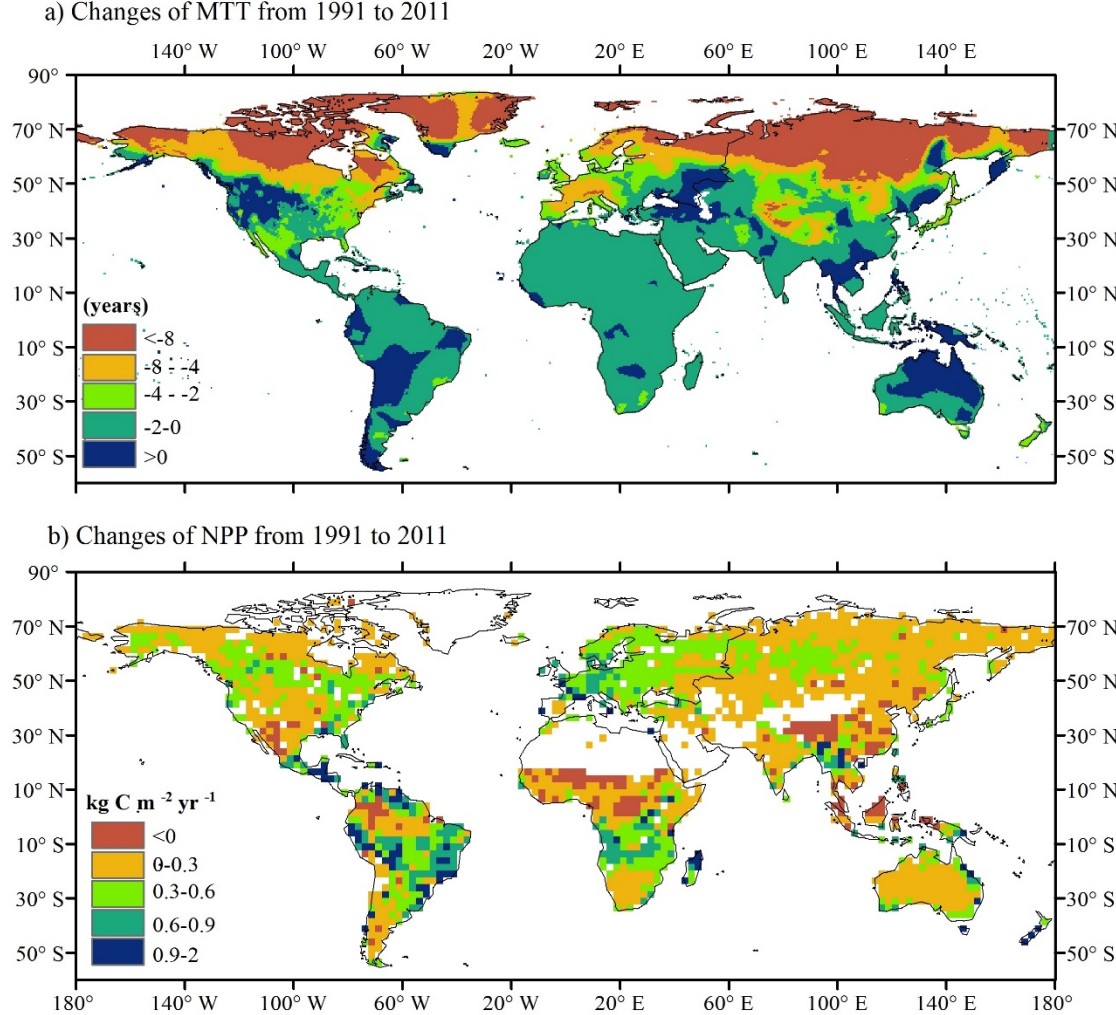

**Figure 7**. Change values of ecosystem mean ecosystem mean turnover time (MTT, unit: year a)

driven by temperature change and NPP (unit: Kg C m$^{-2}$yr$^{-1}$) from 1901 to 2011. MTT for 1901 and 2011

was calculated by the temperature-dependence function showing in Fig. 4. NPP in 1901 and 2011 was

derived from models' average and MODIS.





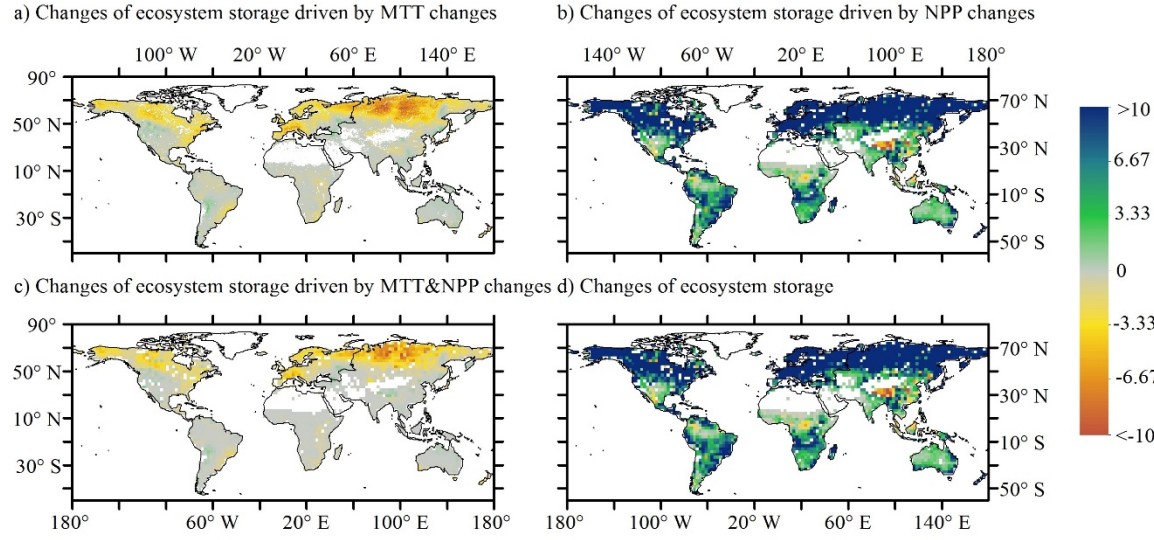

**Figure 8**. Change values of ecosystem carbon storage driven by mean turnover time change

($NPP_{2011} \times \Delta MTT$, a), by NPP change ($MTT_{2011} \times \Delta NPP$, b) and by NPP change and MRT change

($\Delta MTT \times \Delta NPP$, c) and total ecosystem C storage changes (d). Unit: g C m$^{-2}$ yr$^{-1}$ ($\Delta C_{pool} = NPP_{2011} \times$

$\Delta MTT + MTT_{2011} \times \Delta NPP - \Delta NPP \times \Delta MTT$).