# Peer review of "Effects of carbon turnover time on terrestrial ecosystem carbon storage"

_Biogeosciences, 2017_

## Referee Comment (RC1) · Anonymous Referee #1 · 17 Jun 2017

**General comments**

This manuscript describes the calculation of mean carbon turnover time, for both ecosystem and soil pools, based on both GPP and NPP. The introduction does a good job of laying out why this is interesting, given uncertainties surrounding terrestrial ecosystem and climate change, disturbance, etc.

There are however many problems here. First, what exactly is the advance of this study over Carvalhais et al. (2014)? This wasn't clear to me.

Second, this analysis mixes (I believe) spatial and temporal trends, assuming that they're equivalent, but this assumption is never explored or even really discussed.

The steady-state assumption is also troubling. I understand why it may be necessary

at a global scale, but the authors should at least estimate how much bias this might be introducing. For example, there are gridded disturbance and forest age maps available that could be incorporated into such a calculation.

The lack of any clear data availability statement is unacceptable. It's 2017, and I expect all code and data to be included as supplementary info, or (better) posted in a repository. It's not acceptable to produce results from a black box.

The figures should be improved. See comments below.

Finally, while I appreciate the difficulties of writing in a foreign language, the current manuscript is riddled with spelling and grammar mistakes. This is doubly frustrating as I know that the senior author, at least, is fluent in English.

**Specific comments**

1. Line 24: Why "Thus"? Doesn't seem to be logically connected

2. L. 28: "difference"

3. L. 47: "validated" probably not the best word to use here

4. L. 52: "amount of"

5. L. 62-63: Carvalhais et al. (2014) seems like a needed citation here

6. L. 86-87: unclear

7. L. 90: this language is used frequently in the ms. Is ecosystem C storage really "driven" by MRT? I would say that MTT is an emergent property of changes in fluxes; it can't "drive" anything

8. L. 116: by the definition above (pool/flux), it *definitely* would change

9. L. 142: cite R correctly ("citation()"), including version numbers of all packages used

10. L. 166: at the biome level, do you mean?

11. L. 196-201: first, need to note that you're assuming that the current-day *spatial* correlation between temperature and MTT is identical to the *temporal* correlation between these variables. It's not at all obvious this would be true. Second, you're mixing models and remote sensing products; it would be good to document how much divergence these models have from MOD17 in 2011.

12. L. 211-221: are these really results? Aren't these just the GLC database numbers?

13. L. 225-: be consistent in using long/short or high/low or large/small in referring to MTT

14. L. 245: Q10 is 1.95 implies that MTT roughly doubles with a 10 °C increase? That seems nonsensical

15. L. 298-299: can you explain this more?

16. L. 338-340: see comment 7 above re language and causality

17. L. 365-366: is it possible to quantify, even in a back-of-the-envelope kind of way, how much error might be introduced by this assumption? That would be interesting

18. L. 389: but you're not measuring temporal variability (much), except for changes over time in the MOD17 product, right?

19. L. 406-419: this is all duplicative and can be removed

20. L. 421-422: completely inadequate data availability statement. Elevation data?!?

21. Figures generally: maps are pretty but have limited utility. At least of these might be more informative if gives as e.g. Latitude versus MTT plots

22. FIgure 6: not at all useful in my opinion

---

## Referee Comment (RC2) · Anonymous Referee #2 · 28 Jun 2017

This manuscript presents ecosystem carbon turnover times calculated globally using MODIS-based GPP and NPP combined with observation-based datasets of plant biomass, litter, and soil carbon. Contemporary turnover times and their potential to change in response to warming and other climatic factors are an important issue in understanding the earth system. However, I think this paper has some issues with its analysis and interpretation that should be addressed.

One major issue is that it isn't clear what the major new advance was in this analysis compared to previous, similar analyses. This analysis seems very similar to that of Carvalhais et al (2014), which is cited several times in the manuscript. In fact, Carvalhais et al was arguably more comprehensive than this analysis because it included direct comparisons to earth system model simulations. I think there were some new features

in this analysis, such as the inclusion of litter estimates, comparing whole ecosystem vs. soil MTT, and looking at changes over the 20th century, but I think the paper could do a better job of highlighting which things are new and how they changed the results relative to previous, similar studies. If the litter estimates are new, then maybe there could be more discussion of how and why including that pool changed the results relative to previous analyses.

Another issue is the potential for bias in some of the results due to the datasets used for GPP and NPP. While MODIS-derived GPP is constrained by satellite observations, it also depends on assumptions about climatic and environmental factors that affect plant growth and photosynthesis. For example, the efficiency parameter that converts absorbed light into GPP varies with VPD and temperature. MODIS NPP includes maintenance respiration that is calculated based on estimates of plant biomass and a temperature-dependent Q10 relationship. This raises questions about the temperature and moisture relationships shown in Figures 4, 5, and 6, as well as the related estimates of changes in MTT over time. It is difficult to tell how much these relationships are affected by the underlying assumptions of the MODIS NPP algorithm. Since the estimates are not completely measurement based, it is harder to be confident about their meaning.

The estimates of changes in MTT over the 20th century are also problematic because NPP in 1901 was modeled rather than measurement-based. This means that all the changes in NPP from 1901 to 2011 are based on a comparison between average output from several models (1901) to a measurement-based (but partially modeled) estimate (MODIS in 2011). How much of the difference was due to climatic factors that changed over that time period and how much was due to differences between the different sets of NPP estimates? I wonder whether the results in Figure 7b (difference between models in 1901 and MODIS NPP in 2011) would look compared to the change in NPP from the model ensemble between 1901 and 2011. In the end, if models of NPP are being compared, what is the advantage of this MTT approach compared to

just analyzing the change in carbon stocks from the actual model output over time?

The analysis depends on a space-for-time substitution (developing temperature and precipitation relationships based on spatial patterns and assuming they also apply to changes over time). What is the potential for bias in this assumption? Processes like acclimation of microbial respiration to warming or shifts in plant species ranges could make changes over time quite different from those that would be expected from observed spatial patterns.

Comparing GPP and NPP as separate and independent metrics doesn't make much sense since both are derived from the same MODIS product. The difference between GPP and NPP is entirely determined by the assumptions of the MODIS NPP algorithm, so I'm not sure I would expect that distinction to provide much useful information in this type of analysis.

In general, I think the Discussion doesn't say enough about why this analysis is useful compared to existing models and previous analyses. The suggestions given for incorporating these results into earth system models and land models are not very useful because most of these factors (e.g., temperature dependence of turnover rates) are already included in all existing models. I do think that there are some useful outcomes from this type of analysis, but I think the Discussion needs some more interpretation of the specific results in the context of ecological factors rather than general statements about how models should take these results into account.

The manuscript also could use some proofreading for English usage.

Specific comments:

Line 56-57: This analysis generally discusses NPP and mean C turnover time as independent, but they could also be related. For example, faster plant growth could accelerate soil C turnover via priming effects, or there could be correlations between plant growth rates and the longevity of vegetation.

Line 62-63: It seems like Carvalhais et al (2014), which this analysis largely follows, did do a pretty good job of quantifying this spatial variation at global scales.

Line 66-68: Another recent radioisotope paper to cite is He et al (2016)

Line 78-82: This suggests that the main contribution of this paper is comparing different versions of MTT calculations. But it's not really clear later on if that is meant to be the focus or not. The paper is also about changes in MTT over time, but doesn't really connect these two parts together.

Line 165-166: "interpret the quantity as an emergent diagnostic at the ecosystem level": What does this emergent diagnostic actually tell us? There isn't any discussion of how it should be interpreted or what kind of bias would occur as a result of the steady state assumption being violated.

Line 180: The equation for MTT looks like it's fitting a ratio of MAT/MAP, but I think this is actually meant to say either MAT or MAP. It's very confusing the way it's currently written.

Line 214-216: If most of the carbon was in soil, then total ecosystem MTT would be largely determined by soil MTT. What are the implications of this when comparing those two estimates?

Line 220: I would expect permafrost soils to have much larger C stocks in places with very deep organic soils. It's not unusual for deep permafrost to have >100 kgC/m2 (Schuur et al., 2015). Could that lead to bias in these results?

Line 224-225: He et al (2016) used radiocarbon analysis to estimate a mean soil C residence time of about 3000 years, which they found to be consistent with several other published estimates. What explains the 2 order of magnitude difference from the estimates here? Turnover time for tundra also seems very short, given that permafrost soils are known to have been steadily accumulating carbon for thousands of years.

Line 256: It doesn't seem like the increase in R2 was really that significant.

Line 261-262: It would be nice to include a map of temperature changes along with MTT and NPP changes so all driving factors could be compared. Also, why was only temperature and not precipitation included in this part of the analysis, even though both looked like they had significant relationships with MTT?

Line 268 and 271: I think these units should be PgC, not PgC/year

Lines 270-275: This might be a good place to discuss whether the whole ecosystem patterns differed from the soil C patterns if there were any interesting patterns there

Line 293-297: I think a lot more could be said about the ecology behind these results. What features of dominant plant species and soil contributed to these differences? Differences in plant lifetime? Tissue lifetime? Susceptibility to decomposition?

Line 299: Since the ratio of GPP to NPP is entirely determined by the assumptions of the MODIS NPP algorithm, I don't think this result has a lot of real-world meaning.

Line 377-379: Why would this reduce the uncertainties?

Line 381-382: Doesn't aggregating everything to the biome level violate the assumptions behind calculating change in MTT over time? This would suggest that MTT could only change if the spatial extent of different biomes was shifting.

Line 390-391: This would be a good place to discuss alternative soil databases like Hengl et al (2014) - available at soilgrids.org

Line 392-393: This is arguably the primary purpose of all land surface models. They all already consider this.

Line 397-398: All land surface models already include temperature functions that affect pool turnover times.

Line 401-404: Land surface models already include these processes. In general, this whole section about improvements to land models isn't supported by any comparison between this study and actual land model output. Carvalhais et al (2014) did explicitly

compare their MTT results to earth system model simulations, and I don't think it makes sense to discuss these model-related suggestions without doing a similar comparison here.

Line 421-422: Data availability would require putting all the MTT data somewhere that readers can access it.

Figure 1: The colors need to be rescaled, especially for soil C. It's really hard to see anything in that map. Also, the soil C has some obvious artifacts, like the sharp change in soil C on the border between Alaska and Canada. What does this mean for the results? It would also be nice to have a map of NPP here so all the drivers could be seen together.

Figure 2: Since all three of these look about the same, I don't really see the point in including all of them as separate metrics

Figure 4: Panel a: The regression looks like it underestimates the slope of the curve by a lot. Panel d: The exponential fit does not do very well at the high precipitation end. What does this mean for the results?

Figure 7: The titles on the figure say from 1991 to 2011, but the text says it goes from 1901 to 2011.

References:

Hengl T, de Jesus JM, MacMillan RA, Batjes NH, Heuvelink GBM, Ribeiro E, et al. (2014) SoilGrids1km — Global Soil Information Based on Automated Mapping. PLoS ONE 9(8): e105992. https://doi.org/10.1371/journal.pone.0105992

He, Y., S. E. Trumbore, M. S. Torn, J. W. Harden, L. J. S. Vaughn, S. D. Allison, and J. T. Randerson (2016), Radiocarbon constraints imply reduced carbon uptake by soils during the 21st century, Science, 353(6306), 1419–1424, doi:10.1126/science.aad4273.

Schuur, E. A. G. et al. (2015), Climate change and the permafrost carbon feedback,

Nature, 520(7546), 171–179, doi:10.1038/nature14338.

---

## Author Comment (AC1) · 31 Aug 2017

Thanks so much for sending us two reviews on our manuscript "The effects of carbon turnover time on terrestrial ecosystem carbon storage" (ID: bg-2017-183). We are very grateful to the reviewers for their constructive comments and suggestions. Their inputs have helped to improve the paper significantly. We have carefully studied the reviews, and revised our manuscript accordingly. The detail information was described in the attached files.

Please also note the supplement to this comment: https://www.biogeosciences-discuss.net/bg-2017-183/bg-2017-183-AC1-supplement.zip

---

## Author Response (AR1)

Dear Editor,

Thanks so much for sending us two reviews on our manuscript "**The effects of carbon turnover time on terrestrial ecosystem carbon storage**" (ID: bg-2017-183). We are very grateful to the reviewers for their constructive comments and suggested amendments. Their inputs have helped to improve the paper significantly. We have carefully studied the reviews, and revised our manuscript accordingly. As a consequence, our manuscript has been considerably improved.

Here are our detailed responses to the reviews. Please note that the comments are in *italics* followed by our responses in **regular** text. In addition, we marked the changes or revision with the **red text** in the whole revised manuscript.

Yours Sincerely,
Xuhui

Xuhui Zhou

School of Ecological and Environmental Sciences,

East China Normal University,

Dongchuan Road, Shanghai 200062, China

E-mail: xhzhou@des.ecnu.edu.cn

**Response letter to comments (ID: bg-2017-183)**

**Referee #1**

**General comments:**

*First, what exactly is the advance of this study over Carvalhais et al. (2014)? This wasn't clear to me.*

**[Response]** Thanks so much for your comments. In *Carvalhais et al. (2014)*, global C turnover times and its covariation with climate were mainly examined. They also compared global C turnover time calculated by the model results from CIMP5 with those from observed data and showed their trend over latitude. Based on their work, we extended litter C and vegetation C pools from different datasets into ecosystem C storage to estimate ecosystem C turnover time compared to the study of *Carvalhais et al.* (2014). We also focused on the uncertainty from datasets, especially in high latitude (HWSD vs. NCSCD). More importantly, we examined ecosystem C storage over time from changes in C turnover time and/or NPP. In addition, we calculated the GPP-based the NPP-based and soil MTTs to explore their difference and its variability to climate. Therefore, our study advance the understanding of the uncertainty of global C turnover time and ecosystem C storage from C turnover time with updated data. We revised the introduction and discussion to make it clearer for the advance in Lines 83-86, 92-94, 312-319.

*Second, this analysis mixes (I believe) spatial and temporal trends, assuming that they're equivalent, but this assumption is never explored or even really discussed.*

**[Response]** Thanks for your suggestions. In this study, we assumed that the spatial correlation between temperature and MTT is similar to the temporal correlation between these variables. We added this assumption and discussed this caveat in Lines 210-211,447-450.

*The steady-state assumption is also troubling. I understand why it may be necessary at a global scale, but the authors should at least estimate how much bias this might be introducing. For example, there are gridded disturbance and forest age maps available that could be incorporated into such a calculation.*

**[Response]** Thanks so much for your comments. For an ecosystem, a steady state is defined as GPP equals total ecosystem respiration at a reasonable period of time and there is no net change in total standing crop of living and dead biomass. However, maintaining a steady state without change is rare for a long time and ecosystems could be only close to reach the steady state in the short time.

As we know, disturbance and forest age structure will influence large-scale accumulation of biomass, the partitioning of C into pools with different turnover times, and thereby long-term C sequestration and turnover times. In the past decades, most of previous studies have considered the age-related decline in forest growth and simulated the current-age C flux to some degree (Zaehle et al., 2006), which were involved in the gridded data. Therefore, the gridded disturbance and forest age maps can be used to simulate the current-age ecosystem turnover time using models to compare our results, although it has large uncertainty. However, the specific effects of disturbance and forest age on ecosystem C turnover time are difficult to be examined, which was beyond our study. We thus added the discussion of the disturbance and forest age effects on ecosystem C turnover time in the discussion section as well as the caveat of the steady state (Lines 431-435, 415-420).

*The lack of any clear data availability statement is unacceptable. It's 2017, and I expect all code and data to be included as supplementary info, or (better) posted in a repository. It's not acceptable to produce results from a black box.*

**[Response]** Thanks for your suggestions. All of the original data (MOD 17, HWSD, NCSCD, vegetation C production of Gibbs et al (2006) and litter dataset from Holland et al., 2005, climate variables from the Climate Research Unit (CRU_TS 3.20)) are open and shared. We provided full citations for data sources in MS and the download links in the supplemental information.

The download links were as follows:
  **MOD 17**: https://modis.gsfc.nasa.gov/data/dataprod/mod17.php
  **HWSD**: http://eusoils.jrc.ec.europa.eu
  **NCSCD**: http://bolin.su.se/data/ncscd/
  **Vegetation C**: http://cdiac.ess-dive.lbl.gov/epubs/ndp/global_carbon/carbon_documentation.html
  **litter dataset**:https://daac.ornl.gov/VEGETATION/guides/Global_Litter_Carbon_Nutrients.html
  **the Climate Research Unit**: http://www.cru.uea.ac.uk/

*The figures should be improved. See comments below.*

**[Response]** See the response as below.

*Finally, while I appreciate the difficulties of writing in a foreign language, the current manuscript is riddled with spelling and grammar mistakes. This is doubly frustrating as I know that the senior author, at least, is fluent in English.*

**[Response]** We carefully revised the manuscript, especially for the language editing. Meanwhile, we asked a native speaker: Shahla Hosseini Bai, to carefully revise the whole manuscript. Hope our manuscript has been considerably improved.

**Specific comments:**

*Line 24: Why "Thus"? Doesn't seem to be logically connected*

*L. 28: "difference"*

**[Response]** Done as suggested.

*L. 47: "validated" probably not the best word to use here*

**[Response]** We revised "validated" to "evaluated".

*L. 52: "amount of"*

**[Response]** Done as suggested.

*L. 62-63: Carvalhais et al. (2014) seems like a needed citation here*

**[Response]** Done as suggested. We added the citation "Carvalhais *et al*. (2014)" in Lines 79-82.

*L. 86-87: unclear*

**[Response]** Done as suggested.

*L. 90: this language is used frequently in the ms. Is ecosystem C storage really "driven" by MRT? I would say that MTT is an emergent property of changes in fluxes; it can't "drive" anything*

**[Response]** Thanks for your comments. The ecosystem C storage is co-determined by C influx and C turnover time. For example, reduced soil C turnover time resulted in the insignificant net effect of increased atmospheric $CO_2$ on the equilibrium soil carbon storage (van Groenigen et al., 2014). Here, we referred to the changes in ecosystem C storage from the changes in C turnover time as the changes of ecosystem C storage driven by turnover time, compared to the changes in ecosystem C storage driven by C influx.

As suggested, we also used other words instead of "driven", such as ecosystem C storage over time from changes in MTT, the MTT-induced changes in ecosystem C storage and so on to diversely show it.

*L. 116: by the definition above (pool/flux), it definitely would change*

**[Response]** We agreed with your comments. We have carefully discussed the difference between ecosystem and soil C turnover times in the discussion section (Lines 319-321, 332-344).

*L. 142: cite R correctly ("citation()"), including version numbers of all packages used*

**[Response]** Sorry for the mistake. We did not use R software to regrid the spatial resolution. Actually we used ARCGIS 10 (ESRI Inc.) and adopted the regridding method from Todd-Brown *et al*. (2013) to regrid the spatial resolution for C fluxes and pool. We have added the citation: ARCGIS 10 and Brown *et al*. (2013), in Lines 152, 154.

*L. 166: at the biome level, do you mean?*

**[Response]** Sorry for the confusion. We aggregated ecosystem C turnover time and mean annual temperature (MAT, $^{o}$C), mean annual precipitation (MAP, mm) and aridity index (AI) into a biome level.

*L. 196-201: first, need to note that you're assuming that the current-day spatial correlation between temperature and MTT is identical to the temporal correlation between these variables. It's not at all obvious this would be true. Second, you're mixing models and remote sensing products; it would be good to document how much divergence these models have from MOD17 in 2011.*

**[Response]** Thanks for your suggestions. We added this assumption in Lines 210-211, and also discussed its limitation in the discussion section (Lines 447-450) as "We assumed that the current-day spatial correlation between temperature and MTT is identical to the temporal correlation between these variables, although such assumption cannot reflect the processes like acclimation of microbial respiration to warming or shifts in plant species over time".

We used NPP in 2011 from MODIS products and NPP in 1901 from models since there was no MODIS GPP in 1901. Our previous paper (Yan *et al*., 2014) showed that the modeled NPP was near to MODIS-estimated NPP and their difference was mostly less than 0.05 kg C m$^{-2}$ yr$^{-1}$, so we used the average modeled NPP (CanESM2, CCSM4, IPSL-CM5A-LR, IPSL-CM5B-LR, MIROC-ESM, MIROC-ESM-CHEM, NorESM1-M and NorESM1-ME) for NPP in 1901 and assumed the average modeled NPP was similar to MODIS NPP in 1901. The detail information was described in Yan *et al*. (2014).

*L. 211-221: are these really results? Aren't these just the GLC database num- bers?*

**[Response]** The terrestrial C storage was calculated from the global datasets about plant biomass, soil and litter C, which described in the datasets section. GLC database was just used for plant functional types or biome class to aggregate all C into a biome level.

*L. 225-: be consistent in using long/short or high/low or large/small in referring to MTT*

**[Response]** Done as suggested. We used long/short in referring to MTT in the whole manuscript.

*L. 245: $Q_{10}$ is 1.95 implies that MTT roughly doubles with a 10 ∘C increase? That seems nonsensical*

**[Response]** The previous research reported that $Q_{10}$ for soil or other C pool was near to or larger than 2. For example, Sanderman et al (2003) calculated a $Q_{10}$ value of 2.9 for soil C turnover time using eddy flux. Foereid et al (2014) used $Q_{10}$ value of 1.5~2.27 for soil pool and 1.29~1.66 for litter pool due to pool properties. Compared to those data, we thought that our results were reasonable.

*L. 298-299: can you explain this more?*

**[Response]** The mean GPP-based MTT was slightly longer than that from Carvlhais *et al*. (2014, 23 years) with the similar method. The difference may result from two aspects. Firstly, ecosystem C storage in this study was the sum of soil, vegetation and litter C pools, while Carvalhais *et al*. (2014) only considered soil and vegetation C pools.

Secondly, the data source of global vegetation C storage was different with our study from Gibbs (2006) and Carvalhais *et al.* (2014) from a collection of estimates for pan-tropical regions and radar remote-sensing retrievals for northern and temperate forests. We added the more explanations in Lines 312-319.

*L. 338-340: see comment 7 above re language and causality*

**[Response]** In this study, we quantified the changes in ecosystem C storage from 1901 to 2011 and partitioned it into three parts from the changes in NPP, in ecosystem MTT, and in both NPP and MTT (seeing equation 3). Our results showed that the decrease in MTT increased ecosystem C loss over time due to the increase in C decomposition rates, while increased NPP enhanced ecosystem C uptake due to the decrease in $CO_2$ input to atmospheric and the increase of vegetation C stocks. We have revised them in Lines 383-389.

*L. 365-366: is it possible to quantify, even in a back-of-the-envelope kind of way, how much error might be introduced by this assumption? That would be interesting*

**[Response]** Thanks for your comments. It is sure that the large uncertainty will be introduced by the steady-state assumption. Currently, most studies still used this assumption to examine ecosystem C capacity and turnover time. For example, there are Carvalhais *et al* (2014), Zhou *et al.* (2012), and Barrett et al. (2006). However, it is very difficult to quantify the uncertainty. It is a big project. We did not have some good approaches to resolve this problem to date. We thus only discussed the limitation of this assumption in our discussion. (See the above response).

*L. 389: but you're not measuring temporal variability (much), except for changes over time in the MOD17 product, right?*

**[Response]** Sorry for the confusion. We used an exponential equation between ecosystem MTT and temperature ($MTT = ae^{-bMAT}$) to calculate ecosystem MTT in 1901 and 2011 for the temporal variability of MTT. MOD17 product was for NPP changes over time.

*L. 406-419: this is all duplicative and can be removed*

**[Response]** Done as suggested.

*L. 421-422: completely inadequate data availability statement. Elevation data?!?*

**[Response]** Sorry for confusion. We revised it as "All of the original data (MOD 17, HWSD, NCSCD, vegetation C production of Gibbs et al (2006) and litter dataset from Holland et al., 2005, climate variables from the Climate Research Unit (CRU_TS 3.20)) are open and shared. We provided full citations for data sources in MS and the download links in the supplemental information.".

*Figures generally: maps are pretty but have limited utility. At least of these might be more informative if gives as e.g. Latitude versus MTT plots*

**[Response]** Thanks for suggested. We have rescaled color and also added the latitude pattern for Figure 7.

[Figure]

**Figure 7.** Altered ecosystem carbon storage due to changes in mean turnover time (MTT, NPP2011×ΔMTT, a), net primary production (NPP, MTT2011×ΔNPP, b), and interaction of NPP and MTT (ΔMTT×ΔNPP, c). Panels d and e are total altered ecosystem C storage changes due to changes in MTT, NPP, and MTT×NPP and their latitudinal gradients from panels a-d, respectively. Unit: g C m$^{-2}$ yr$^{-1}$ ($\Delta C_{pool} = NPP_{2011} \times \Delta MTT + MTT_{2011} \times \Delta NPP - \Delta NPP \times \Delta MTT$).

*Figure 6: not at all useful in my opinion*

. **[Response]** Done as suggested. We deleted it in the revised version.

**Referee #2**

**General comments:**

*One major issue is that it isn't clear what the major new advance was in this analysis compared to previous, similar analyses. This analysis seems very similar to that of Carvalhais et al (2014), which is cited several times in the manuscript. In fact, Carvalhais et al was arguably more comprehensive than this analysis because it included direct comparisons to earth system model simulations. I think there were some new features in this analysis, such as the inclusion of litter estimates, comparing whole ecosystem vs. soil MTT, and looking at changes over the 20th century, but I think the paper could do a better job of highlighting which things are new and how they changed the results relative to previous, similar studies. If the litter estimates are new, then maybe there could be more discussion of how and why including that pool changed the results relative to previous analyses.*

[Response] Thanks for your comments and suggestions. In *Carvalhais et al. (2014)*, global C turnover times and its covariation with climate were mainly examined. They also compared global C turnover time calculated by the model results from CIMP5 with those from observed data and showed their trend over latitude. Based on their work, we extended litter C and vegetation C pools from different datasets into ecosystem C storage to estimate ecosystem C turnover time compared to the study of *Carvalhais et al. (2014)*. We also focused on the uncertainty from datasets, especially in high latitude (HWSD vs. NCSCD). More importantly, we examined the changes in ecosystem C storage over time from changes in C turnover time and/or NPP. In addition, we calculated the GPP-based, the NPP-based and soil MTTs to explore their difference and its variability to climate. Therefore, our study advances the understanding of the uncertainty of global C turnover time and ecosystem C storage from C turnover time with updated data. We revised the introduction and discussion to make it clearer for the advance in Lines 83-86, 92-94, 312-319.

*Another issue is the potential for bias in some of the results due to the datasets used for GPP and NPP. While MODIS-derived GPP is constrained by satellite observations, it also depends on assumptions about climatic and environmental factors that affect plant growth and photosynthesis. For example, the efficiency parameter that converts absorbed light into GPP varies with VPD and temperature. MODIS NPP includes maintenance respiration that is calculated based on estimates of plant biomass and a temperature-dependent Q10 relationship. This raises questions about the temperature*

*and moisture relationships shown in Figures 4, 5, and 6, as well as the related estimates of changes in MTT over time. It is difficult to tell how much these relationships are affected by the underlying assumptions of the MODIS NPP algorithm. Since the estimates are not completely measurement based, it is harder to be confident about their meaning.*

**[Response]** Thank for your comments. Mean turnover time (MTT) was calculated as the ratio of C storage and C influx (e.g., GPP or NPP), so the relationships could be affected by the relationships of GPP or NPP with temperature and VPD. The MODIS NPP algorithm would affect the estimates of MTT, which were discussed in the previous paper (Zhao et al., 2005; Zhao, M. and Running, S. W. 2010) but the uncertainty was within the allowable range. We thus thought that the uncertainty from the underlying assumptions of MODIS NPP algorithm was not considered in this study.

Specifically, the MDOIS NPP algorithm is expressed as: $NPP = \sum_{i=1}^{365} PsnNet - (R_{mo} + R_g)$, where PsnNet is net photosynthesis ($PsnNet = GPP - R_{ml} - R_{mr}$). $R_{ml}$ and $R_{mr}$ are maintenance respiration by leaves and fine roots, respectively. $R_{mo}$ is maintenance respiration by all other living parts except leaves and fine roots (e.g., livewood), and $R_g$ is growth respiration. GPP was calculated as: $GPP = \varepsilon * FPAR * PAR$, where $\varepsilon$ is the radiation use efficiency of the vegetation determined by maximum $\varepsilon$ in each biome and temperature and soil moisture. All the parameters were abstained from the MOD17 Biome Parameter Look-Up Table (BPLUT). Therefore, the performance of the algorithm can be largely influenced by algorithm itself as well as the uncertainties from upstream inputs, such as land cover, FPAR/LAI, the meteorological data. For C5 MOD17, the BPLUT and the upstream inputs were be improved, so the MOD17 NPP is comparable to the Ecosystem Model–Data Intercomparison (EMDI) NPP data set, and global total MODIS GPP and NPP are inversely related to the observed atmospheric $CO_2$ growth rates, and MEI index, indicating that MOD17 are reliable products (Zhao et al., 2005). For example, direct comparison of MODIS annual GPP (MOD17A3) with observations for 37 site-years has resulted in a higher correlation and lower bias ($r^2=0.6993$, relative error=19%, unpublished data) than MODIS annual GPP calculated using tower meteorology ($r^2=0.595$, relative error=2%).

*The estimates of changes in MTT over the 20th century are also problematic because NPP in 1901 was modeled rather than measurement-based. This means that all the changes in NPP from 1901 to 2011 are based on a comparison between average output from several models (1901) to a measurement-based (but partially modeled) estimate*

*(MODIS in 2011). How much of the difference was due to climatic factors that changed over that time period and how much was due to differences between the different sets of NPP estimates? I wonder whether the results in Figure 7b (difference between models in 1901 and MODIS NPP in 2011) would look compared to the change in NPP from the model ensemble between 1901 and 2011. In the end, if models of NPP are being compared, what is the advantage of this MTT approach compared to just analyzing the change in carbon stocks from the actual model output over time?*

**[Response]** Thank for your comments. We used NPP in 2011 from MODIS products and NPP in 1901 from models, which were no MODIS GPP in 1901. Our previous paper (Yan *et al*., 2014) showed that the modeled NPP was near to MODIS NPP and their difference was mostly less than 0.05 kg C m$^{-2}$ yr$^{-1}$. We thus used the average modeled NPP (CanESM2, CCSM4, IPSL-CM5A-LR, IPSL-CM5B-LR, MIROC-ESM, MIROC-ESM-CHEM, NorESM1-M and NorESM1-ME) for NPP in 1901 and assumed that the average model NPP was similar to MODIS NPP in 1901. Since the details had been described in Yan *et al*., (2014), we did not add the detailed comparison of between NPP of model ensemble in 1901 and MODIS NPP in 2011 and NPP from the model ensemble between 1901 and 2011 in this study. To clarify the information, we added the relevant description and references in Lines 212-216, 622.

*The analysis depends on a space-for-time substitution (developing temperature and precipitation relationships based on spatial patterns and assuming they also apply to changes over time). What is the potential for bias in this assumption? Processes like acclimation of microbial respiration to warming or shifts in plant species ranges could make changes over time quite different from those that would be expected from observed spatial patterns.*

**[Response]** Thank for your comments. We assumed that the current-day spatial correlation between temperature and MTT is identical to the temporal correlation between these variables in the revised MS, because there is no time series of data between MTT and temperature at the global scale. However, such assumption cannot reflect the processes like acclimation of microbial respiration to warming or shifts in plant species ranges as suggested by the reviewer, which could make changes over time. In the revised MS, we added the limitation for this assumption in the discussion section (Lines 447-450).

*Comparing GPP and NPP as separate and independent metrics doesn't make much*

*sense since both are derived from the same MODIS product. The difference between*
*GPP and NPP is entirely determined by the assumptions of the MODIS NPP algorithm,*
*so I'm not sure I would expect that distinction to provide much useful information in this*
*type of analysis.*

**[Response]** Thanks so much for your comments and suggestions. Thompson and Randerson et al (1999) has indicated that there were two types of mean C turnover times for terrestrial ecosystems: the GPP-based or the NPP-based mean turnover time according to the terrestrial C models with GPP or NPP as their C inputs, respectively (i.e., NPP is GPP minus autotrophic respiration). However, there was no clear distinction in most pervious researches, so we calculated the both MTTs for comparison. In addition, NPP-based MTT is more available in comparison with soil MTT than GPP-based MTT in the literature. The difference between GPP-based and NPP-based MTT was determined by the ratio of GPP and NPP, which largely influenced by the assumptions of the MODIS NPP algorithm. We added their difference in the discussion sections (Lines 319-321).

*In general, I think the Discussion doesn't say enough about why this analysis is useful compared to existing models and previous analyses. The suggestions given for incorporating these results into earth system models and land models are not very useful because most of these factors (e.g., temperature dependence of turnover rates) are already included in all existing models. I do think that there are some useful outcomes from this type of analysis, but I think the Discussion needs some more interpretation of the specific results in the context of ecological factors rather than general statements about how models should take these results into account.*

**[Response]** Thanks for your comments and suggestions. In the Discussion section, we added two examples (e.g., the lifetime and decomposition) to the context of ecological factors in the revised MS as "The difference between NPP-based ecosystem and soil MTT was the turnover time of vegetation and litter, which was related to plant functional types (PFTs). For instance, the difference between NPP-based and soil MTTs in Australia was small (33.4 and 29.8 years, respectively) compared to that in other regions, because one of the PFTs accounting for a large space of Australia was spare grass with short turnover time (3.5 years on average). In addition, within a specific PFT, different ecosystems may have diverse turnover time due to climate effects. NPP-based and soil MTTs for boreal neadleaf evergreen forest were about 116 years and 98 years, respectively, while both for tropical ones were about 12 years and 8 years, although ecosystem C in boreal and tropic zone was in the same order of magnitude (~34 vs. 40

kg C m-2) with the similar vegetation C storage (~3.5 kg C m-2). High temperature and humidity in tropical zone, which promote decomposition processes, contribute to the short turnover time compared to those in boreal zone (Sanderman et al., 2003)." in Lines 332-344.

*The manuscript also could use some proofreading for English usage.*

**[Response]** We carefully revised the manuscript, especially for the language editing. Meanwhile, we asked a native speaker: Shahla Hosseini Bai, to carefully revise the manuscript. Hope our manuscript has been considerably improved.

**Specific comments***:*

*Line 56-57: This analysis generally discusses NPP and mean C turnover time as independent, but they could also be related. For example, faster plant growth could accelerate soil C turnover via priming effects, or there could be correlations between plant growth rates and the longevity of vegetation.*

**[Response]** Thanks for comments. The transient C storage is determined by the MTT and NPP. If climate increases C influx (NPP) into an ecosystem but does not change C transient times (MTT), the C sequestration rate of the ecosystem increases due to the fact that more C stays in the ecosystem for the same length of time, which could be correlated between growth C plant growth rates and the longevity of vegetation. Certainty, climate would increases C influx and also accelerate soil C turnover, so the C sequestration rate of the ecosystem increases, which is determined by both the amounts of C influx and their MTT. Therefore, in this study, we firstly partitioned the changes in the C storage into three parts: from the changes in NPP, MTT and both (seeing equation 3), and secondly, the NPP and MTT in 1901 and 2011 were used to estimate the changes in ecosystem C storage over time, and finally we discussed the spatial pattern of ecosystem C storage changes and the possible reasons.

*Line 62-63: It seems like Carvalhais et al (2014), which this analysis largely follows, did do a pretty good job of quantifying this spatial variation at global scales.*

**[Response]** Thanks for your comments. Based on their works (Carvalhais et al (2014)), we extended litter C and vegetation C pools from different datasets into ecosystem C storage to estimate C turnover time and evaluate their uncertainty from datasets. We mainly focused on comparing different versions of MTTs and quantifying the spatial variation in ecosystem C storage over time from the changes in C turnover time and/or C flux.

*Line 66-68: Another recent radioisotope paper to cite is He et al (2016)*

**[Response]** Thanks you for providing the new reference.

*Line 78-82: This suggests that the main contribution of this paper is comparing different versions of MTT calculations. But it's not really clear later on if that is meant to be the focus or not. The paper is also about changes in MTT over time, but doesn't really connect these two parts together.*

**[Response]** In this study, we focused on comparing different versions of MTT and its effects to climate as well as quantifying the changes in ecosystem C storage due to ecosystem MTT. The changes in MTT over time was used to estimate the ecosystem C storage changes caused by MTT (equ. 3), which was calculated by the relationship between MTT and climate.

*Line 165-166: "interpret the quantity as an emergent diagnostic at the ecosystem level": What does this emergent diagnostic actually tell us? There isn't any discussion of how it should be interpreted or what kind of bias would occur as a result of the steady state assumption being violated.*

**[Response]** If the ecosystem is not at the steady state, the C turnover time cannot be calculated by the ratio of C storage and C flux. We thus followed the assumption of Carvalhais *et al.* (2014) and have discussed the limitation of steady-state in MS (Lines 415-420).

*Line 180: The equation for MTT looks like it's fitting a ratio of MAT/MAP, but I think this is actually meant to say either MAT or MAP. It's very confusing the way it's currently written.*

**[Response]** Done as suggested. We revised it as $\text{MTT} = a e^{-b MAT \text{ or } MAP}$.

*Line 214-216: If most of the carbon was in soil, then total ecosystem MTT would be largely determined by soil MTT. What are the implications of this when comparing those two estimates?*

**[Response]** $\text{MTT}_{EC} = \frac{Cpool}{GPP} = \frac{C_{soil+}C_{veg}+C_{litter}}{NPP/\varepsilon} = \varepsilon * MTT_{soil} + \varepsilon * \frac{C_{veg}+C_{litter}}{NPP}$ ($\varepsilon = \frac{NPP}{GPP}, MTT_{soil} = \frac{C_{soil}}{NPP}$). If most of the C was in soil, the ratio of NPP to GPP is the key to determine the difference between GPP-based ecosystem MTT and soil MTT and NPP-based MTT is similar to soil MTT. We have discussed the difference versions of MTTs in Lines 319-321, 332-344.

*Line 220: I would expect permafrost soils to have much larger C stocks in places with very deep organic soils. It's not unusual for deep permafrost to have >100 kgC/m2 (Schuur et al., 2015). Could that lead to bias in these results?*

**[Response]** In this study, ecosystem MTT was calculated as the ratio of C storage and influx ($\text{MTT}_{EC} = \frac{Cpool}{GPP} = \frac{C_{soil+}C_{veg}+C_{litter}}{GPP}$). When the deep permafrost is considered, the ecosystem MTT would become longer. If we assumed that soil C in deep permafrost is 100 kg C/m$^2$ and GPP is 0.2 kg C m$^2$ yr$^{-1}$, the MTT is 500 years.

*Line 224-225: He et al (2016) used radiocarbon analysis to estimate a mean soil C residence time of about 3000 years, which they found to be consistent with several other published estimates. What explains the 2 order of magnitude difference from the estimates here? Turnover time for tundra also seems very short, given that permafrost soils are known to have been steadily accumulating carbon for thousands of years.*

**[Response]** In our MS or Carvalhais et al. (2014), we assumed that ecosystem was in the steady state and calculated MTT as the ratio of C pool and C flux. Here, we did not separate C pools into fast, slow, or passive, which could largely underestimate the ecosystem MTT. Another factor is that the current soil dataset such as HWSD underestimate the soil C storage, especially for permafrost soils. In the discussion section, we discussed the limitation of the assumption of the steady-state and the difference soil datasets effects on the estimate of ecosystem turnover time (Lines 415-420, 421-429).

*Line 256: It doesn't seem like the increase in R2 was really that significant.*

**[Response]** Sorry for the confusion. $R^2$ for the regression function of soil MTT with MAT was 0.76 when AI>1, while $R^2$ was 0.52 when AI<1 (Fig. 5e & h)

*Line 261-262: It would be nice to include a map of temperature changes along with*

*MTT and NPP changes so all driving factors could be compared. Also, why was only temperature and not precipitation included in this part of the analysis, even though both looked like they had significant relationships with MTT?*

**[Response]** Done as suggested. We added a map of temperature changes in Figure 7 (Figure 6 in the revised MS, seeing the below). There is no change in $R^2$ when MAP was incorporated into the regression function of ecosystem MTT with MAT, so we just considered the temperature changes included in this part of the analysis.

*Line 268 and 271: I think these units should be PgC, not PgC/year*

**[Response]** Sorry for the mistake. Thanks so much for your correlation. This unit is Pg C for the change of C storage from 2011 to 1901. We have revised it in Lines 293, 296.

*Lines 270-275: This might be a good place to discuss whether the whole ecosystem patterns differed from the soil C patterns if there were any interesting patterns there*

**[Response]** Thanks for your suggestions. Patterns of ecosystem and soil C storage can determined by NPP and MTT, so the difference between the whole ecosystem and the soil C patterns was determined by the difference between ecosystem and soil MTT. In our study, MTT in 1901 and 2011 was calculated using the relationship between MTT and temperature, so the difference of temperature functions determined the difference of both MTT and then the C storage patterns. Therefore, we added the limitation of MTT calculation in the discussion section (Lines 436-447). When the relationship between soil MTT and temperature was used ($MTT_{soil} = 58.40e^{-0.08MAT}$), the changes on ecosystem C storage caused by MTT could decrease 161.42 Pg C and that driven by NPP could be 1125.6 Pg C, with the similar spatial pattern as the ecosystem.

*Line 293-297: I think a lot more could be said about the ecology behind these results. What features of dominant plant species and soil contributed to these differences? Differences in plant lifetime? Tissue lifetime? Susceptibility to decomposition?*

**[Response]** Thanks so much for suggestions. The difference between NPP-based ecosystem and soil MTT was the turnover time of vegetation and litter, which was related to plant functional types (PFTs). We have added some ecological information behind these results (e.g., plant lifetime, decomposition as suggested) in Lines 332-344

as "The difference between NPP-based ecosystem and soil MTT was the turnover time of vegetation and litter, which was related to plant functional types (PFTs). For instance, the difference between NPP-based and soil MTTs in Australia was small (33.4 and 29.8 years, respectively) compared to that in other regions, because one of the PFTs accounting for a large space of Australia was spare grass with short turnover time (3.5 years on average). In addition, within a specific PFT, different ecosystems may have diverse turnover time due to climate effects. NPP-based and soil MTTs for boreal neadleaf evergreen forest were about 116 years and 98 years, respectively, while both for tropical ones were about 12 years and 8 years, although ecosystem C in boreal and tropic zone was in the same order of magnitude (~34 vs. 40 kg C m-2) with the similar vegetation C storage (~3.5 kg C m-2). High temperature and humidity in tropical zone, which promote decomposition processes, contribute to the short turnover time compared to those in boreal zone (Sanderman et al., 2003).".

*Line 299: Since the ratio of GPP to NPP is entirely determined by the assumptions of the MODIS NPP algorithm, I don't think this result has a lot of real-world meaning.*

**[Response]** Two types of mean C turnover times has been suggested for terrestrial ecosystems: the GPP-based or the NPP-based mean turnover time according to the terrestrial C models with GPP or NPP as their C inputs, respectively (Thompson and Randerson *et al*., 1999, NPP is GPP minus plant respiration). However, there was no clear distinction in most pervious researches, so we calculated the both two for comparison and NPP-based MT is more available in comparison with soil MTT than GPP-based MTT. In our study, the difference between GPP-based and NPP-based MTT was determined by the ratio of GPP and NPP, which largely influenced by the assumptions of the MODIS NPP algorithm. We added these information in Lines 319-321.

*Line 377-379: Why would this reduce the uncertainties?*

**[Response]** Sorry for the confusion. We deleted this sentence in the revised version after we carefully considered the sources of uncertainties.

*Line 381-382: Doesn't aggregating everything to the biome level violate the assumptions behind calculating change in MTT over time? This would suggest that MTT could only change if the spatial extent of different biomes was shifting.*

**[Response]** The original data, including MOD 17, HWSD, vegetation C production of Gibbs *et al.* (2006), and litter dataset from Holland *et al.* (2005), were created based on the plant functional types (PFTs) or biomes by the assumptions of algorithm, so we aggregated MTT into a biome level to estimate the change in MTT over time for data match.

*Line 390-391: This would be a good place to discuss alternative soil databases like Hengl et al (2014) - available at soilgrids.org*

**[Response]** Thanks for your suggestions. We have discussed the uncertainty caused by the different datasets (in Lines 347-349, 425-429) and also added the soil databases of Hengl et al (2014). (If SoilGrids (Hengl et al., 2014) was used, the MTT in the top 1 m could increase to 30.3 years for GPP-based, 66.9 years for NPP-based and 45.7 years for soil. )

*Line 392-393: This is arguably the primary purpose of all land surface models. They all already consider this.*

**[Response]** Thanks for your comments. We have deleted it.

*Line 397-398: All land surface models already include temperature functions that affect pool turnover times.*

**[Response]** Thanks for your comments. It is sure that all land surface models already included temperature functions that affected C pools and fluxes via plant photosynthesis and respiration. These effects probably directly affected turnover times of C pools to some degree. Carvalhais *et al.* (2014) examined the covariation of climate with turnover times. In this study, we emphasized the effects of moisture or precipitation on soil decomposition, especially in high-latitude zones with greater warming and increased precipitation.

*Line 401-404: Land surface models already include these processes. In general, this whole section about improvements to land models isn't supported by any comparison between this study and actual land model output. Carvalhais et al (2014) did explicitly compare their MTT results to earth system model simulations, and I don't think it makes sense to discuss these model-related suggestions without doing a similar comparison here.*

**[Response]** Thanks for your comments and suggestions. Compared with Carvalhais *et al*. (2014), we mainly discussed the difference of the climate effects between on ecosystem MTT and soil MTT, especially for moisture. Our results also showed that the temperature sensitivity of ecosystem turnover time was lower than that of soil C pool ($Q_{10}$: 1.95 vs. 2.23), while moisture stress on soil MTT was significant, especially under low aridity conditions. Current land surface models have considered moisture stress on vegetation, but the incorporation of moisture or precipitation stress into soil decomposition should be strengthened, especially in high-latitude zones with greater warming and increased precipitation.   To make it clear, we have rewrote these sentences (Lines 461-469).

*Line 421-422: Data availability would require putting all the MTT data somewhere that readers can access it.*

**[Response]** All of the original data (MOD 17, HWSD, NCSCD, vegetation C production of Gibbs et al (2006) and litter dataset from Holland et al., 2005, climate variables from the Climate Research Unit (CRU_TS 3.20)) are open and shared. We provided full citations for data sources in MS and the download links in the supplemental information.

The download links were as follows:

**MOD 17**: https://modis.gsfc.nasa.gov/data/dataprod/mod17.php

**HWSD**: http://eusoils.jrc.ec.europa.eu
**NCSCD**: http://bolin.su.se/data/ncscd/
**Vegetation C**: http://cdiac.ess-dive.lbl.gov/epubs/ndp/global_carbon/carbon_ documentation.html
**litter dataset**:  https://daac.ornl.gov/VEGETATION/guides/Global_Litter_Carbon_ Nutrients.html
**the Climate Research Unit**: http://www.cru.uea.ac.uk/

*Figure 1: The colors need to be rescaled, especially for soil C. It's really hard to see anything in that map. Also, the soil C has some obvious artifacts, like the sharp change in soil C on the border between Alaska and Canada. What does this mean for the results? It would also be nice to have a map of NPP here s ons.*

**[Response]** Done as suggested. We have rescaled the colors for map (Figure 1).

Soil C storage in Alaska is near 30 kg C, while that in Canada is less than 10 kg C, forming the sharp change in soil MTT on the border between Alaska and Canada. Here, soil MTT in Alaska ranges among 70~95 years and that in Canada on the border is less than 20 years.

Figure 1 showed the C storage in different C pools (soil, vegetation, litter and ecosystem). Since NPP is not our focus in this study, we put NPP map in Supplemental information.

[Figure]

*Figure 2: Since all three of these look about the same, I don't really see the point in including all of them as separate metrics*

**[Response]**Thanks for your comments. The colors have been rescaled to strengthen

[Figure]

the difference among three of these.

*Figure 4: Panel a: The regression looks like it underestimates the slope of the curve by a lot. Panel d: The exponential fit does not do very well at the high precipitation end. What does this mean for the results?*

**[Response]** We agree that the curve fit does not do very well at the high precipitation end at Panel a or Panel b. If the high precipitation (>2000mm) was neglected, the exponential fit would do better. For example, $R^2$ would increase to 0.86 at Panel d.

*Figure 7: The titles on the figure say from 1991 to 2011, but the text says it goes from 1901 to 2011.*

**[Response]** We have revised the titles on the figure from 1901 to 2011.

[Figure]

NPP

is GPP minus  plant respiration).

In addition,  soil C

turnover time are usually estimated using field sampling as the global turnover time for model validation. However, the difference among different versions of turnover time ere still unclear.

Therefore,

We  calculated  the GPP-based  NPP-based ecosystem and soil turnover time through the similar method to explore the difference

C turnover time and/or NPP. Therefore, our study advance the understanding of the uncertainty of global C turnover time (especially in high latitude) and ecosystem C storage from C turnover time with updated data. and its effectsvariability to climates.

we examined compared the difference of turnover times between both and also consideredbased on soil, 
[revised manuscript text omitted]

**4   Discussion**

In Carvalhais et al. (2014), global C turnover times and its covariation with climate were mainly examined. They also compared global C turnover time calculated by the model results from CIMP5 with those from observed data as well as their trend over latitude. Based on their work, we focused on the uncertainty from different observed data (HWSD vs. NCSCD), especially in high latitude. Litter data was updated compared to the study of Carvalhais et al.

(2014). We also estimated the GPP-based the NPP-based and soil MTT to explore the difference among them. More importantly, we examined ecosystem C storage over time from changes in C turnover time and/or NPP. Therefore, our study advance the understanding of the uncertainty of global C turnover time (especially in high latitude) and ecosystem C

storage from C turnover time with updated data.

4.1 Global pattern of mean turnover time

In this study, we used the ratio of C storage to C flux to calculate the GPP-based, the NPP- based and soil MTT, and compared their difference. we estimated spatial patterns of mean turnover time (MTT) with ecosystem C influxes (GPP and NPP) and C pools in plants, litter

Terrestrial ecosystems comprise of compartments varying greatly in their individual turnover times (e.g., leaves, wood, different soil organic carbon fractions), but we cannot estimate turnover time for each pools using observation datasets. In addition, it is difficult to accurately get the observed respiration ($R_a$ and $R_h$) in terrestrial ecosystem at the global scale, or carbon C allocation between outflux and influx. It is thus difficult to evaluate how this assumption affects model results. Maybe, inverse models would be a valid method to estimate turnover time for the both (e.g., Zhou *et al.*, 2012).

The global average of ecosystem MTT was 25.0 years for GPP-based estimation and 50.8

years for NPP-based one, and soil MTT was 35.5 years, which were within the global mean turnover times (26-60 years) estimated by various experimental and modeling approaches with NPP-based estimation (Randerson *et al.*, 1999; Thompson and Randerson,

1999)

between ecosystem and soil MTT depends on the component carbon pools and the ratio of

GPP to NPP. Thus, there was subtle difference in patterns of MTT between both. For example, ecosystem MTT in Evergreen Needleleaf forest (ENF) was larger than soil MTT

where the decomposition rate in soil C was very slow. The mean GPP-based MTT are was slightly longer than the result ofthat from Carvalhais *et al.* (2014) (, 23 years) with the similar method. The difference may result from two aspects. There are two possible factors explaining their difference. Firstly, ecosystem C storage in this study was the sum of the soil, vegetation and litter C pools, while Carvalhais *et al.* (2014) justonly considered the soil and vegetation C pools. Secondly, the data source of global vegetation C storage was different with our study from global vegetation C storage came from the result of Gibbs (2006) and , while Carvalhais *et al.* (2014) usedfrom 
[revised manuscript text omitted]

$\Delta MTT$).

**Figure 8**. Change values of ecosystem carbon storage driven by mean turnover time change (NPP$_{2011}$×∆MTT, a), by NPP change (MTT$_{2011}$×∆NPP, b) and by NPP change and

MRT change (∆MTT×∆NPP, c) and total ecosystem C storage changes (d). Unit: g C m$^{-2}$ yr

$^{-1}$ ($\Delta C_{pool} = NPP_{2011} \times \Delta MTT + MTT_{2011} \times \Delta NPP - \Delta NPP \times \Delta MTT$).

---

## Author Response (AR2)

Dear Editor,

Thanks so much for sending us the reviews on our manuscript "**The effects of carbon turnover time on terrestrial ecosystem carbon storage**" (ID: bg-2017-183). We are very grateful to the reviewers for their constructive comments and suggested amendments. We have carefully studied the reviews, and revised our manuscript accordingly. We also asked a native speaker Shahla Hosseini Bai, to carefully revise the manuscript.

Here are our detailed responses to the reviews. Please note that the comments are in *italics* followed by our responses in **regular** text. In addition, we marked the changes or revision with the **red text** in the whole revised manuscript.

Yours Sincerely,

Xuhui

Xuhui Zhou

School of Ecological and Environmental Sciences,

East China Normal University,

500 Dongchuan Road, Shanghai 200062, China

E-mail: xhzhou@des.ecnu.edu.cn

**Response letter to comments (ID: bg-2017-183)**

**Referee #1**

**General comments:**

*There do remain a few problems: terms need to be carefully defined, some improvements for readability would be useful, and the ms still needs careful editing for English usage throughout–see specific comments below.*

**[Response]** We carefully defined the terms again, and asked a native speaker Shahla Hosseini Bai, to carefully revise the manuscript again. Hope the language has been considerably improved.

*Specific comments*

*1. Line 21: need to define NPP and GPP*

*2. L. 33: "deserve further"*

**[Response]** Done as suggested.

*3. L. 77: probably start new paragraph*

**[Response]** Thanks for your suggestions. This paragraph showed the main methods to estimate the C turnover time and their pros and cons. Since the ratio of C storage to flux is one of the methods for C turnover time estimates, we did not start new paragraph.

*4. L. 97-102: not sure this should be here; delete or move to methods?*

**[Response]** Done as suggested. We deleted it.

*5. L. 167-169: for the future, note Hashimoto et al. datasets:*

*http://cse.ffpri.affrc.go.jp/shojih/data/index.html*

**[Response]** Thanks for providing the data link about global gridded soil respiration. We deleted the sentence "However, it is difficult to accurately get the observed respiration ($R_a$ and $R_h$) in terrestrial ecosystem at the global scale." in the revised MS.

We would consider the soil respiration into the estimate of C turnover time in the future.

*6. L. 222: again, if you use R, please cite it correctly*

**[Response]** Done as suggested. We added the specific citation in revised MS

("through Markov chain Monte Carlo (MCMC) sampling from a gamma distribution

(CRAN: MCMCpack, Martin, et al., 2011)").

Martin, A.D., Kevin, M. Q., and Jong, H.P. 2011. MCMCpack: Markov chain monte

carlo in R. J. Stat. Softw. 42.

*7. L. 247: "longest…shortest"*

*8. L. 419: "and perhaps catastrophically"*

*9. L. 453: be more positive! "Our results provide insights…"*

**[Response]** Done as suggested.